# Exploring the impact of atmospheric forcing and basal drag on the Antarctic ice sheet under Last Glacial Maximum conditions

Javier Blasco[1,2], Jorge Alvarez-Solas[1,2], Alexander Robinson[1,2,3], and Marisa Montoya[1,2]

[1]Departamento de Física de la Tierra y Astrofísica, Universidad Complutense de Madrid, Facultad de Ciencias Físicas, 28040 Madrid, Spain
[2]Instituto de Geociencias, Consejo Superior de Investigaciones Científícas-Universidad Complutense de Madrid, 28040 Madrid, Spain
[3]Potsdam Institute for Climate Impact Research, 14473 Potsdam, Germany

*Correspondence to:* J. Blasco (jablasco@ucm.es)

**Abstract.** Little is known about the distribution of ice in the Antarctic ice sheet (AIS) during the Last Glacial Maximum (LGM). Whereas marine and terrestrial geological data indicate that the grounded ice advanced to a position close to the continental-shelf break, the total ice volume is unclear. Glacial boundary conditions are potentially important sources of uncertainty, in particular basal friction and climatic boundary conditions. Basal friction exerts a strong control on the large-scale dynamics of the ice sheet and thus affects its size, and is not well constrained. Glacial climatic boundary conditions determine the net accumulation and ice temperature, and are also poorly known. Here we explore the effect of the uncertainty in both features on the total simulated ice storage of the AIS at the LGM. For this purpose we use a hybrid ice-sheet-shelf model that is forced with different basal-drag choices and glacial background climatic conditions obtained from the LGM ensemble climate simulations of the third phase of the Paleoclimate Modelling Intercomparison Project (PMIP3). Overall, we find that the spread in the simulated ice volume for the tested basal drag parameterisations is about the same range as for the differente GCM forcings (4 to 6 m sea level equivalent). For a wide range of plausible basal friction configurations, the simulated ice dynamics vary widely but all simulations produce fully extended ice sheets towards the continental-shelf break. More dynamically active ice sheets correspond to lower ice volumes, while they remain consistent with the available constraints on ice extent. Thus, this work points to the possibility of an AIS with very active ice streams during the LGM. In addition, we find that the surface boundary temperature field plays a crucial role in determining the ice extent through its effect on viscosity. For ice sheets of a similar extent and comparable dynamics, we find that the precipitation field determines the total AIS volume. However, precipitation is highly uncertain. Climatic fields simulated by climate models show more precipitation in coastal regions than a spatially uniform anomaly, which can lead to larger ice volumes. Our results strongly support using these paleoclimatic fields to simulate and study the LGM and potentially other time periods like the Last Interglacial. However, their accuracy must be assessed as well, as differences between climate model forcing leads to a large spread in the simulated ice volume and extension.

# 1 Introduction

Sea-level variations on long timescales are driven by the waxing and waning of large continental ice sheets. The characterisation of the sensitivity of ice sheets to past climate changes is fundamental to gaining insight into their underlying dynamics as well as their response to future climate change. In addition, understanding past sea-level changes is important for quantifying sea-level rise (Nicholls and Cazenave, 2010; Defrance et al., 2017; King and Harrington, 2018; Golledge et al., 2019; Robel et al., 2019) and for assessing the risk of crossing tipping points within the Earth System, such as the collapse of the West Antarctic Ice Sheet (Kopp et al., 2009; Sutter et al., 2016; Pattyn et al., 2018).

The Antarctic Ice Sheet (AIS), in particular, plays a fundamental role as it is the largest ice sheet on Earth and stores ca. 58 meters of sea-level equivalent (msle; Fretwell et al. (2013)). Due to its size it is potentially the largest contributor to future sea-level projections, but it is also the most uncertain (Collins et al., 2013). Assessing the AIS contribution to the total sea-level budget at different time periods has proven to be challenging. The Last Glacial Maximum (LGM, 21 ka BP) represents an ideal benchmark period since there is a large availability and variety of proxy data that, furthermore, indicate important AIS changes relative to present day (PD). Both, marine and terrestrial geological data, indicate that at the LGM, the AIS extended to the continental-shelf break (Anderson et al., 2002, 2014; Hillenbrand et al., 2012, 2014; The RAISED Consortium, 2014; Mackintosh et al., 2014). However, its exact extent is not well constrained everywhere. Whereas its advance in the Amundsen region, the Bellingshausen Sea and the Antarctic Peninsula is well established, in the Ross Sea and the East Antarctic region it remains controversial (Stolldorf et al., 2012; The RAISED Consortium, 2014). Furthermore, the total AIS ice volume is even less well constrained (Simms et al. (2019) and references therein). Geological data furthermore do not provide direct information on past thickness and volume of ice sheets, which must hence be inferred. There have been several approaches to infer past ice-volume change of an individual ice sheet as the AIS. One approach is to use direct ice-sheet modelling to simulate the volume of the AIS at the LGM (e.g Huybrechts (2002); Whitehouse et al. (2012a); Golledge et al. (2012); Gomez et al. (2013); Maris et al. (2014); Briggs et al. (2014); Quiquet et al. (2018)). An alternative is to use Glacial Isostatic Adjustment (GIA) modelling, which describes the viscous response of the solid Earth to past changes in surface loading by ice and water (e.g. Ivins and James (2005); Bassett et al. (2007)). This approach has also been used in combination with direct ice-sheet modelling (e.g. Whitehouse et al. (2012b)) and/or by making use of constraints on ice-thickness from reconstructions based on exposure age dating, as well as satellite observations of current uplift (Whitehouse et al., 2012b; Ivins et al., 2013; Argus et al., 2014b). Whereas older studies estimated large sea-level contributions generally above 15 m (e.g. Nakada et al. (2000); Huybrechts (2002); Peltier and Fairbanks (2006); Philippon et al. (2006); Bassett et al. (2007)), more recent modelling studies and reconstructions have lowered these estimates to 7.5-13.5 m (Mackintosh et al., 2011; Whitehouse et al., 2012a; Golledge et al., 2012, 2014; Gomez et al., 2013; Argus et al., 2014b; Briggs et al., 2014; Maris et al., 2014; Sutter et al., 2019). This lowering in ice volume can be explained by the fact that the first ice-sheet models were based purely on the Shallow Ice Approximation for inland ice. This solution solves for slow moving ice, based on shear deformation. However, later models include more sophisticated approximations (e.g. Shallow Shelf Approximation, Full Stokes) with a better representation of fast flowing ice streams. These fast flowing regions contribute to a decrease in ice volume. Nevertheless, the latest LGM AIS

volume estimates still differ by more than 5 m. Part of this difference can be explained by spatial resolution and sub-grid scale grounding-line treatment (e.g. Goelzer et al. (2017); Pattyn (2018)). Other possible explanations include the implementation of external processes, like the GIA (e.g., Whitehouse et al. (2019)), or, as this work, the effect of uncertain climatologies and ice-sheet dynamics.

Ablation rates at the PD are almost zero except for localized areas (van Wessem et al., 2016, 2018). Because the LGM is a colder period, around 10 ℃ as shown by ice core records (Jouzel et al., 2007), ablation rates in the LGM would have been probably negligible. On the other hand, basal melting rates from the LGM are difficult to estimate due to the scarcity of oceanic temperature reconstructions. Nonetheless, geomorphological records point to a fully advanced AIS during the LGM (The RAISED Consortium, 2014). This could hint to low basal-melting rates inside the continental-shelf break. Therefore
ice-sheet dynamics and accumulation must have been the two main factors controlling ice-mass gain during this period. The representation of ice dynamics in ice-sheet models is a key feature that can potentially lead to important discrepancies. Most ice-sheet models simulating the past long-term evolution of large-scale ice sheets are hybrid models that rely on the Shallow Ice Approximation (SIA) and the Shallow Shelf Approximation (SSA). Moreover, there is no universally accepted friction law, and basal friction is treated in different manners in ice-sheet models. Ritz et al. (2015) emphasize the importance of the basal
friction, as it can favour the occurrence of the marine instability in future AIS projections. Generally, basal stress follows either a power-law formulation on the basal ice velocity (a special case being the Weertman (1957) friction law) or a Coulomb friction law (Schoof, 2005) with different power-law coefficients, a friction coefficient and potentially a regularization term. Ice-sheet models thus use friction formulations that can range from linear viscous and regularized Coulomb friction laws, typical of hard bedrock sliding (Larour et al., 2012; Pattyn et al., 2013; Joughin et al., 2019) to Coulomb-plastic deformation, characteristic
of ice flow over a soft bedrock with filled cavities (Schoof, 2005, 2006; Nowicki et al., 2013). In the simplest cases a constant friction coefficient is prescribed over the whole domain (Golledge et al., 2012), but generally this parameter incorporates the dependency of basal friction on the effective pressure exerted by the ice, as well as on bedrock characteristics by making use of assumed till properties (Winkelmann et al., 2011; Albrecht et al., 2019; Sutter et al., 2019) or basal temperature conditions (Pattyn, 2017; Quiquet et al., 2018). The sensitivity of the simulated ice volume to these features is substantial. For instance,
Briggs et al. (2013) obtained differences of more than 5 msle for an Antarctic LGM state depending only on the friction coefficients used for hard and soft beds. Some studies have attempted to overcome the uncertainty in basal friction by optimising the friction coefficient through inversion methods in order to obtain an accurate PD ice-sheet state (Morlighem et al., 2013; Le clec'h et al., 2019). However, these optimizations are based on a particular configuration of the PD state, and it is unclear whether they remain valid for glacial conditions. All in all, basal friction is poorly characterised, and the potential consequences
of the associated uncertainty should be considered in ice-sheet modeling.

    Glacial atmospheric boundary conditions over Antarctica are also far from being well constrained. It is clear from ice-core records and marine deep-sea sediment data that, at the continental scale, temperatures were lower than today and that the climate was drier (Frieler et al., 2015; Fudge et al., 2016). Typically, ice-sheet models use two approaches for simulating the atmospheric conditions at the LGM. On one hand, some studies prescribe a spatially-uniform temperature anomaly (generally
between 8 K and 10 K below PD) and a uniform reduction in precipitation (generally by 40-50% compared to PD), as inferred

from individual ice-core records (Huybrechts, 2002; Golledge et al., 2012; Whitehouse et al., 2012a; Gomez et al., 2013; Quiquet et al., 2018). However, this approach provides only a crude representation of glacial climate anomalies. In reality, even if ice cores show a similar temperature decrease, estimated precipitation changes are less homogeneous. Thus imposing a constant change over the whole domain will potentially misrepresent climatologies in localized areas (Frieler et al., 2015; Fudge et al., 2016). In addition, ice cores are extracted from domes and the recorded changes are not necessarily representative of coastal regions. Because the LGM is a cold state, with presumably no (or negligible) ablation and oceanic basal melt, the reduction of precipitation with respect to the PD should have an important impact on the size of the simulated ice sheet. In addition, because the temperature and/or precipitation anomalies are uniform, the PD pattern is imprinted on the LGM atmospheric forcing fields, and changes in atmospheric patterns are thus neglected.

Another commonly used method is to prescribe the LGM temperature and precipitation fields for the whole Antarctic domain from climate simulations (Briggs et al., 2013; Maris et al., 2014; Sutter et al., 2019). Output from simulations using a hierarchy of climate models has been used in the literature, from global general circulation models (GCMs) (Sutter et al., 2019), sometimes downscaled with regional models (Maris et al., 2014), to Earth System Models of Intermediate Complexity (EMICs) (Blasco et al., 2019). Briggs et al. (2013) went a step forward to investigate the effect of uncertainty in the climate forcing fields by assessing the effect of the inter-model variance through an empirical orthogonal function (EOF) analysis. However, some model outputs do not simulate the temperature anomalies correctly at specific sites where proxies are available, such as Vostok or Dome C. This may lead to an unrealistic configuration and thus it is necessary to evaluate the accuracy of model outputs (Cauquoin et al., 2015).

In this work we aim to assess the effects of the uncertainty in basal friction and climatic (in particular atmospheric) boundary conditions on the simulated LGM AIS. We focus on basal-drag choices which can lead to realistic LGM states. For these we then investigate the effect of different temperature and precipitation fields. To this end, we use a thermomechanical ice-sheet-shelf model forced with LGM background conditions. The atmospheric temperature and precipitation fields are obtained from the eleven GCMs participating in the Paleoclimate Modelling Intercomparison Project Phase III (PMIP3) as part of the Coupled Model Intercomparison Project Phase 5 (CMIP5, Taylor et al. (2012)). The article is structured as follows. First, we describe the ice-sheet-shelf model used and the experimental setup (Section 2). Then, we show the results obtained for different basal friction coefficients and atmospheric conditions (Section 3). Finally, the results are discussed (Section 4) and summarized in the conclusions (Section 5).

## 2   Methods and experimental setup

For this study we use the three-dimensional, hybrid, thermomechanical ice-sheet-shelf model Yelmo (Robinson et al., 2020). The model covers the whole Antarctic domain with 191x191 grid cells of 32 km x 32 km resolution and 21 layers in sigma-coordinates. The flow of the grounded ice is computed as the sum of the solutions of the Shallow Ice Approximation (SIA, Hutter (1983)) and the Shallow Shelf Approximation (SSA, MacAyeal (1989)). Sliding occurs only within the SSA solution, where the computed basal velocity is modulated with the corresponding basal friction. Ice shelves are solved within the SSA

solution without basal drag. The initial topographic conditions (ice thickness, surface and bedrock elevation) are obtained from the RTopo-2 dataset (Schaffer et al., 2016). The internal ice temperature is calculated via the advection-diffusion equation.

Yelmo computes the total mass balance (MB) as a sum of the surface mass balance (SMB), the basal mass balance at the ice base and calving at the ice front. The SMB is obtained from the difference between the ice accumulation through precipitation and surface melting using the positive degree-day method (PDD; Reeh (1989)). Although there are more comprehensive methods that account for short-wave radiation for instance (Robinson et al., 2011), the PDD scheme is commonly used in ice models in the Antarctic domain, because ablation at these latitudes is limited (Winkelmann et al., 2011; Pollard and DeConto, 2012; Pattyn, 2017). Furthermore, in this particular study, the transient character of the AIS evolution is not simulated, as we focus on the LGM period. Thus, there is no need to explicitly account for the effects of changes in insolation on melting. Calving occurs when the ice-front thickness decreases below an imposed threshold (200 m in this study) and the upstream ice flux is not large enough to provide the necessary ice for maintaining the previous thickness (Peyaud et al., 2007). Present-day basal melting rates at the ice-shelf base and at the grounding line are obtained from Rignot et al. (2013) and extrapolated over all 27 basins identified by Zwally et al. (2012). Below grounded ice, the basal mass balance is determined through the heat equation as in Greve and Blatter (2009), where the geothermal heat flux field is obtained from Shapiro and Ritzwoller (2004). The glacial isostatic adjustment (GIA) is computed with the elastic lithosphere-relaxed asthenosphere (ELRA) method (Le Meur and Huybrechts, 1996), where the relaxation time of the asthenosphere is set to 3000 years.

Yelmo does not explicitly model the impact of ice anisotropy on the ice flow, so an "enhancement factor" is used as a tuning parameter (Ma et al., 2010; Pollard and DeConto, 2012; Maris et al., 2014; Albrecht et al., 2019). For this study we found realistic PD states for $E_{grounded}=1.0$ and for ice shelves $E_{floating}=0.7$.

## 2.1 Basal-drag law

As mentioned above basal sliding is calculated within the SSA solution, which is a function of the basal stress. Yelmo computes the basal stress at the ice base ($\boldsymbol{\tau_b}$) through a linear viscous friction law. It depends on the basal ice velocity ($\mathbf{u_b}$), the effective ice pressure ($N_{eff}$) and a tunable friction coefficient ($c_b$):

$$\boldsymbol{\tau_b} = \beta \mathbf{u_b}, \tag{1}$$

and

$$\beta = c_b N_{eff} \tag{2}$$

is the basal-drag coefficient, in [kPa yr m$^{-1}$]. $c_b$, given in [yr m$^{-1}$], is a coefficient that reflects the bedrock characteristics, and $N_{eff}$ is the effective ice pressure, given in [kPa]. Here we have parameterized $c_b$ as a function of the bedrock elevation, $z_b$ (positive above sea level), analogous to previous work (e.g., Martin et al. (2011)):

$$c_b = \begin{cases} c_{max} & \text{if } z_b \geq 0 \\ \max\left[ c_{max} \exp\left(-\frac{z_b}{z_0}\right), c_{min} \right] & \text{if } z_b < 0 \end{cases} \tag{3}$$

Here, $z_0$ is an internal parameter that determines the bedrock e-folding depth over which the friction coefficient $c_b$ decreases from a maximum value of $c_{\max}$ reached for bedrock elevations above sea level ($z_b \geq 0$) and a minimum threshold value $c_{\min}$. For lower values of $z_0$, $c_b$ falls more rapidly with depth. This parameterisation captures the phenomenon by which the occurrence of sliding (and its intensity) is favoured at low bedrock elevations and specifically within the marine sectors of ice sheets. It follows a similar approach as in Albrecht et al. (2019) and Martin et al. (2011), where the bedrock friction (in their case the "till friction angle") depends on the bedrock elevation.

The effective pressure is represented by the Leguy et al. (2014) formulation, under the assumption that the subglacial drainage system is hydrologically well connected to the ocean so that there is full support from the ocean wherever the ice-sheet base is below sea level. We thus assume that the exerted basal pressure at the land-ice interface depends on the difference between the overburden pressure and the basal water pressure (i.e. the distance from flotation as measured in ice thickness), hence:

$$N_{\text{eff}} = \rho_i g \left( H - H_f \right) \tag{4}$$

where $\rho_i$ is the density of ice, g is gravity, H is the ice thickness and $H_f$ is the flotation thickness, given by $H_f = \max\left[ 0, -\frac{\rho_w}{\rho_i} z_b \right]$, where $\rho_w$ is the seawater density, respectively, and $z_b$ is the bedrock elevation (positive above sea-level). In this way, far from the grounding line, $H_f = 0$ and $N_{\text{eff}} = \rho_i g H$, while at the grounding line, where $H = H_f$, $N_{\text{eff}} = 0$. This ensures continuity of $\tau_{\mathbf{b}}$ at the grounding line.

## 2.2  Climate forcing

To simulate the AIS at the LGM, Yelmo is run over 80 kyr with constant LGM conditions from PD observations. Sea level was set at -120 m during the LGM. The atmospheric forcing field is given by the following equation:

$$T_{\text{LGM}}^{atm} = T_0^{atm} + \Delta T_{\text{LGM-PD}}^{atm} \tag{5}$$

where $T_0^{atm}$ is the PD temperature field at sea level obtained from RACMO2.3 forced by the ERA-Interim reanalysis data (Van Wessem et al., 2014) and $\Delta T_{\text{LGM-PD}}^{atm}$ is the LGM surface temperature anomaly relative to the PD. The monthly-mean temperature fields are obtained from each of the the eleven PMIP3 models, as well as by the ensemble mean (Fig. 1a). We apply a lapse rate correction that accounts for changes in elevation (0.008 K m$^{-1}$ for annual temperatures and 0.0065 K m$^{-1}$ for summer temperatures) in concordance with other ice-sheet models (Ritz et al., 1997; DeConto and Pollard, 2016; Quiquet et al., 2018; Albrecht et al., 2019).

The LGM precipitation is calculated as

$$P_{\text{LGM}} = P_0 \delta P_{\text{LGM/PD}} \tag{6}$$

where $P_0$ is the PD monthly-mean precipitation obtained in the same way as the PD temperature and $\delta P_{\text{LGM/PD}}$ is the relative anomaly between the LGM and PD obtained from the PMIP3 ensemble. Figure 1b shows the resulting precipitation field, $P_{\text{LGM}}$, for the PMIP3 ensemble mean. Precipitation is corrected with local temperature anomalies through Clausius-Clapeyron

scaling which assumes more accumulation for warmer temperatures and therefore lower elevations (5 %K$^{-1}$; Frieler et al. 2015). Note that precipitation is given in water equivalent and transformed into accumulation via changes in density (i.e. 1 m yr$^{-1}$ water equivalent ca. 1.09 m ice). Basal-melting rates for floating ice shelves are set to zero in the LGM state for simplicity.

## 2.3 Experimental set-up of the sensitivity studies

**Basal friction**

To investigate the impact of changes in basal friction on the LGM AIS we assess the sensitivity to the friction in marine zones via the minimum friction allowed ($c_{min}$) and the elevation parameter ($z_0$) in Eq. 3 that controls how quickly friction decreases with depth. For this purpose we force Yelmo with a single reference climatic state obtained from the average anomaly of the PMIP3 ensemble for the LGM climate (Fig. 1) and a range of friction parameters. This range was determined in two steps. First, PD AIS simulations were carried out. Values of $c_{max}$= 200·10$^{-5}$ yr m$^{-1}$ were found to simulate the PD AIS in good agreement with observations in terms of grounded ice volume and grounding-line advance for the selected range of values of $c_{min}$ = 1·10$^{-5}$, 3·10$^{-5}$ and 5·10$^{-5}$ yr m$^{-1}$ and of $z_0$ = -100, -125, -150, -175 and -200 m (Fig. 2; see Supplementary Information, Fig. S1, S2 for 2D-snapshots). The parameter range for the LGM AIS simulations was then selected under the criterion that the simulated volume of ice above flotation in the corresponding PD AIS simulation is within ±1 msle of that calculated from PD observations as in Schaffer et al. (2016) (grey band in Fig.2).

**Climatic fields**

To understand the impact of changes in climatic forcing on the ice sheet, we fix the friction parameter values to a single, reference set of values which simulate the best PD state (Fig. 3, $z_0$ = -150 m and $c_{min}$ = 5·10$^{-5}$ yr m$^{-1}$) and analyze the AIS simulated at the LGM for the climatic forcing derived from each of the 11 models in the PMIP3 ensemble, using the aforementioned forcings for temperature (Eq. 5) and precipitation (Eq. 6). We focus on how the temperature and precipitation fields control the size and extent of the ice sheet. In all experiments the sea-level change estimates are computed with respect to the simulated PD state for the reference friction parameter values.

## 3 Results

## 3.1 Impact of basal friction

Here we present the simulated AIS equilibrium configuration under LGM conditions for different basal friction parameters. Ice volume change is converted into a sea-level contribution by subtracting the floating portion and taking isostatic depression of the bedrock into account (Goelzer et al., 2019). Figure 4a shows how the simulated ice volume (in msle) varies with the mean basal-drag coefficient ($\beta$) of the marine zones for $c_{min}$ = 1·10$^{-5}$ yr m$^{-1}$ (circles), 3·10$^{-5}$ yr m$^{-1}$ (crosses) and 5·10$^{-5}$ yr m$^{-1}$ (diamonds) (SM, Fig. S3 for individual snapshots and Fig. S4 for time evolution). A higher mean marine friction

(associated with lower $z_0$ values) is found to result in a larger ice volume. Sea-level differences between a case with rapidly decreasing marine friction (e.g. $z_0$=-100 m; in red) and a case with more gradually decreasing friction (e.g. $z_0$=-200 m, in blue) are about 7 msle. This can be explained by the fact that basal friction reduces basal sliding and hence the ice flow, translating into thicker ice. Faster sliding in the deepest areas (lowest $c_{min}$ values) also reduces ice volume, by about 5 msle for the range

of parameters explored. We do not identify a strong impact of marine basal friction on equilibrium grounded ice area, as the final grounding line configuration is similar in all ensemble members (Fig. 4b). However, as discussed later, this can be due to the long integration time (SM, Fig. S4). Our results fit well within the range of previous studies both in terms of simulated msle (Simms et al. (2019) and references therein) and reconstructions of ice extension from ICE-6G (Argus et al., 2014a; Peltier et al., 2015, 2018), The RAISED Consortium (2014) and the ANU reconstruction (Lambeck and Johnston, 1998; Lambeck

and Chappell, 2001; Lambeck et al., 2002, 2003). Note that in order to avoid biases due to Yelmo's coarse spatial resolution, these extensions were computed using the ice-sheet margins of each of the reconstructions at Yelmo's spatial resolution (SM, Fig. S5). For the simulations that matched PD AIS volumes within $\pm 1$ msle to observations, LGM ice volumes differences between 12.3 to 15.1 msle and ice extension about 16 million $km^2$ were computed.

Looking at the simulated ice thickness between the LGM and the PD state we find a similar pattern for a slowly decreasing

basal friction ($z_0$=-200 m; Fig. 5b) and a more rapidly decreasing friction ($z_0$=-150 m; Fig. 5a). The main source of the LGM volume difference comes primary from the WAIS, especially from the Ross and Ronne shelf, as they advanced up to the continental-shelf break. Also a slight ice thickness decrease is found in the center of the EAIS. Performing an anomaly study between these two states allows to analyze the effect of the employed basal friction parameterisation (Fig. 5c). Ice volume differences primarily originate in the WAIS and the coastal marine regions of the EAIS and its surroundings. This occurs as a

consequence of ice streams which become faster on topographic lows, such as the Amery, Wilkes and Victorias Land (Fig. 5d) leading to thinner ice. These zones of fast flowing areas are similar to the predicted occurrence of basal sliding from (Golledge et al., 2012).

Subtle differences are found when comparing the extension of grounded ice in our simulated AIS with previous reconstructions. Our simulated grounded area covers almost 16 million $km^2$ of the 17 million $km^2$ of the continental-shelf break (i.e.

defined by the contour $z_b$=-2000 m). Our simulated extension stands between the ICE-6G model and the RAISED Consortium and the ANU model. The largest discrepancies between models occur on the Ross shelf (SM, Fig. S5). Whereas ANU and RAISED estimate an advance close to the continental-shelf break, ICE-6G is more retreated, while our results support a nearly complete advance except for $z_0$=-200 m and $c_{min}$ =$5 \cdot 10^{-5}$ yr m$^{-1}$.

## 3.2 Impact of climatic forcing

Here we present the simulated LGM AIS of each individual PMIP3 model for the reference friction parameters (Fig. 6) (SM, Fig. S6 for time evolution and Fig. S7 for velocity distribution). The simulated ice-volume anomaly ranges from 9.6 msle to 15.4 msle (Fig. 7), a spread of 5.8 msle. We excluded in this range the model CNRM-CM5, which we will discuss later. The total ice extension ranges from 15.9 million $km^2$ to 14.6 million $km^2$ , a difference of 1.3 million $km^2$. Thus, while the

spread in ice volume is somewhat smaller than found when investigating the sensitivity to friction, the spread in extension is significantly larger.

Because the underlying dynamics in Yelmo are the same in all cases, the differences in size and extension can only be explained by differences in the climatic fields. To determine the causes underlying these differences, we investigate the sensitivity of the ice thickness and extension to the climatic fields used to force the ice-sheet model (Fig. 8). We find that higher accumulation results in a thicker ice sheet (Fig. 8a), but has no strong effect on the ice extension (Fig. 8b). For model climatologies for which the LGM ice sheet extends close to the continental-shelf break (an extension of around 15.5 million $km^2$, see Fig 8d), the AIS ice volume increases with increasing accumulation (Fig. 8c). However, there are four climate models (CNRM-CM5, GISS-E2-R-150, GISS-E2-R-151, FGOALS-g2) that despite having higher accumulation on average than the ensemble mean, do not allow the ice sheet to advance as much as the other models, leading in all cases to extensions below 15 million $km^2$ (Fig. 8b). Therefore, the simulated AIS volume is smaller for these less advanced ice sheets, despite the relatively high accumulation rates imposed. For all the others, for which extension is around 15.5 million $km^2$, the AIS ice volume clearly increases with increasing accumulation (Fig. 8c).

Further inspection allows us to identify the surface temperature close to the grounding line (Fig 8d) as a critical factor in determining how far the AIS advances. The grounding-line temperature is defined as the mean temperature of the ice column for all the grounding-line grid points. Whereas low surface temperatures lead to similar ice extend,relatively warm surface temperature forcing results in smaller equilibrium grounding line advance. Given the overall low surface temperature at LGM, ablation can generally be discarded as the source of this behaviour (SI Fig. S8; there is, however, one exception, as discussed below and a small area of ablation rates in the Antarctic Peninsula for GISS models), so we turn our attention to ice viscosity. A necessary condition for marine-based ice sheets to advance is that the ice thickness at the grounding line overcomes the flotation criterion as sustained through accumulation and/or by inland ice flow. This condition is fulfilled when the ocean depth ($z_b$) is shallower than ∼90% of the ice thickness. Warmer ice temperatures lower the ice viscosity (Fig. 8e) and prevent the grounding-line from thickening, as a consequence of enhanced ice flow, and advance towards more depressed bedrock zones. Therefore, simulations with lower ice viscosity such as GISS-E2-R-150, GISS-E2-R-151 and FGOALS-g2 do not fully advance in the Ross shelf, Pine Island or the Amery Through (Fig. 6,7).

The CNRM-CM5 model simulates the smallest AIS LGM for all the PMIP3 models. This model expands partly at the Ross shelf and Antarctic Peninsula zone, but collapses completely in the Ronne and Amery shelf, leading to ice free zones in the EAIS and a lower ice volume than the PD (Fig. 6). This occurs due to the presence of ablation in these regions (see SI, Fig. S8). Such a configuration is highly unlikely compared with sea-level and ice extension reconstructions from the LGM. We will discuss later possible explanations for this behaviour.

In summary, we find that the choice of the boundary climate conditions is crucial for the simulated LGM ice sheet. On one hand, the atmospheric temperatures near the coastal regions control the ice extension through viscosity. If the viscosity is low, then the ice flows too fast, preventing the necessary thickening for advancing towards the continental-shelf break. Particularly, if the bedrock is too deep, the ice sheet's expansion will be hampered. Secondly, if the ice sheet extends close to the continental-

shelf break, then the accumulation pattern will determine the total amount of ice volume. We find that for fully extended ice sheets (IPSL-CM5A-LR and MRI-CGCM3), the sea-level difference due to accumulation differences is about 4.2 msle.

**Spatially homogeneous approach**

Applying a simple scheme that lowers the ice accumulation and surface temperature homogeneously over the whole domain
is a common approach at first order, because during the LGM, at continental scale, a colder and drier climate is expected (Huybrechts, 2002; Golledge et al., 2012; Whitehouse et al., 2012a; Gomez et al., 2013; Quiquet et al., 2018). We thus tested a spatially homogeneous scaling (hereafter, the homogeneous method) for comparison. All simulations produce SLE ice volume in the range of previous studies and ice extensions similar to reconstructions (e.g. The RAISED Consortium 2014) if using the same coefficients for basal friction and different climate forcings. Overall, consistently lower ice volumes are simulated with
the homogeneous method, up to 1.5 msle (except for one case SM, Fig. S8). This is solely due to the difference in forcing, as the parameterisation of ice flow is identical. Fig. 9c illustrates the ice thickness difference between the two methods for a similar ice extension (Fig. 9a,b). It is evident that the main source of ice volume differences is due to changes in the WAIS configuration. The Antarctic Peninsula in particular shows a high positive thickness anomaly for the average PMIP3 climatic fields relative to the homogeneous case. In the EAIS, the anomalies are not so pronounced; however, inland ice is slightly thinner, whereas
closer to the coast it is thicker. This anomaly pattern can be explained by the difference between the accumulation fields (Fig. 9d). The spatially homogeneous method accumulates more ice inland and and leads to a reduced accumulation towards the continental-shelf break, especially at the Ross shelf, Pine Island and the Antarctic Peninsula. Because ice cores are generally extracted from dome regions with colder conditions, it is expected that precipitation and air temperatures near the coast are underestimated by the homogeneous approach. Nonetheless, the grounding-line is slightly more advanced in the western region
of the Antarctic Peninsula. Similar as with the different PMIP3 fields, we argue that this difference is due to changes in viscosity due to atmospheric temperatures (SM, Fig. S8).

## 4   Discussion

### 4.1   Steady-state simulations

In this study we assumed steady-state LGM and PD conditions to investigate the effect of climatological boundary conditions
and basal drag parameterisation. Of course, this represents a simplification of reality, as full LGM conditions only occurred for a couple of millennia. In a transient simulation, the results would additionally include a potential internal drift, which we tried to avoid. Although simulations were forced during 80 kyr under steady LGM conditions, equilibrated states were reached after only 30 to 40 kyr (see SM, Fig. S4, S6). Given that the LGP was a cold and sufficiently long period in the Antarctic domain, constant LGM conditions should be enough to stabilize the AIS near its real LGM state.
The simulated PD configurations show a slightly more advanced grounding line in the WAIS compared to the observations, especially at the Ronne shelf (Fig. 3, SM Fig. S1, S2). Also the ice thickness in the interior of the WAIS is systematically lower

than observations. Both features can be partially explained by the basal-drag parameterisation used. Our parameterisation enhances sliding for deeper bedrock. The WAIS is in its vast majority a marine ice sheet, where bedrock depths can reach up to 2000 m below sea level in the interior regions. Thus we systematically simulate a lower WAIS, as we overestimate the ice flow at the interior. This, in addition, promotes the grounding line to advance. Nonetheless, this parameterisation allows

for a precise tracing of ice streams. Except in the Larsen embayment, ice shelves generally show a slightly larger extension than observations. Because larger ice shelves allow for more ice accumulation and exert a backward force, it also helps the grounding-line to advance. Thus, the more advanced grounding line in the Ronne, Amundsen sea and Amery shelves could be additionally explained by the backward force exerted by ice shelves. Nonetheless, the overall picture of the simulated AIS fits well with observations in terms of grounding-line position as well as simulated ice volumes.

**4.2 Role of basal friction**

Even at present-day it is difficult to estimate bed properties like basal temperature or ice velocities, which could improve our understanding of basal friction. Therefore, estimating bed properties at the LGM, where the total ice volume and extension is not fully constrained, adds a degree of difficulty. The dynamical state of the LGM remains a source of uncertainty as there are no observations from that time period of the AIS configuration. To study potentially possible AIS LGM dynamical states,

we covered a range of friction values which lead to realistic LGM and PD configurations. The simulated sea-level differences were about 4 msle between the end members (Fig. 4). We found that the choice of different bedrock frictions has an impact on ice-stream activity in marine-based regions. For example, an AIS that extends up to the continental-shelf break, but with a relatively low volume increase, can be achieved through a very dynamically active ice sheet. In that case, marine-based regions, and more specifically the WAIS, have the potential to maintain fast ice streams at the LGM.

The choice of the friction law for the whole AIS is still somewhat arbitrary and unconstrained. We focused on a linear viscous friction law commonly used in other studies (Morlighem et al., 2013; Quiquet et al., 2018; Alvarez-Solas et al., 2019). We are aware that other types of friction laws could have been tested, such as a regularized Coulomb law (Joughin et al., 2019) or a Coulomb-plastic behaviour (Nowicki et al., 2013), typically for ice flowing over a bedrock filled with cavities. However, the aim of this work was to study the uncertainty associated with the basal drag parameters, rather than assessing the uncertainty

for different friction laws. Given the large uncertainty we quantified for only one friction formulation, we expect that this range would increase further considering additional formulations.

**4.3 Sea-level and ice extent uncertainty**

For our reference friction parameters we used the individual climate simulations of the participating PMIP3 groups as surface boundary forcing. The sea-level difference between the models was about 5.8 msle. The lowest sea-level contribution was 9.6

msle (CCSM4, with exception of CNRM-CM5) and the largest 15.2 msle (IPSL-CM5A-LR). These sea-level estimates were inside the range of other studies and reconstructions. From this point of view, we were not able to discard any specific model field.

The CNRM-CM5 model is a particular model which simulates lower sea-level contributions than PD and more retreated grounding-lines in the Ronne sector and zones of the EAIS. The model CNRM-CM5 simulates the warmest LGM temperatures not only in the SH, but it has been also shown to simulate the lowest LGM volumes for the NH (Niu et al., 2019). A potential explanation for this behaviour can be due to sea-ice formation. As shown in Marzocchi and Jansen (2017), the CNRM-CM5 model simulates the lowest austral sea-ice extent. Such a low extent would increase surface temperatures through sea-ice albedo feedback. Hence, this could point to sea-ice formation as a crucial element in driving fully LGM conditions.

The simulated grounding line advance is strongly influenced by air temperature. Warmer temperatures lower the ice viscosity. Due to the marine character of the AIS, a lower viscosity enhances ice flow leading to thin ice in regions where the bedrock is too deep, which prevents a complete advance towards the continental-shelf break. Forcing from the models CCSM4, FGOALS-g2, GISS-E2-R-150 and GISS-E2-R-151 for instance do not allow a full advance in the Ross shelf, resembling the ICE-6G reconstruction (Fig. 6). On the other hand, if temperatures are sufficiently cold (<-20 °C) ice full advances as in the ANU reconstruction (SM, Fig. S4). The RAISED Consortium has a similar extension, but presents retreated areas at the margins of the Ronne shelf, which we are not able to simulate. Again, the simulated ice extensions were inside the range of the reconstructions, and we could not exclude any case. But we found that in addition to the precipitation field, temperature fields play a crucial role as they have the potential to accelerate the ice by lowering the viscosity and determine the total grounded ice area, which in turn affects the grounded ice volume.

In this study, no basal melting was considered during the LGM. Of course, this is a vast simplification of reality. Unfortunately, reconstructions of ocean subsurface temperatures at the LGM are not available, so that the geological evidence for basal melt is lacking. As shown in Golledge et al. (2012), oceanic forcing leads to a dynamic response of LGM ice streams in the WAIS. If basal melt would have been considered, this would have most likely reduced the total LGM ice volume and affected its extension. Thus, our results represent an upper limit which would reduce when oceanic forcing is considered.

From the point of view of modelling, there have been some attempts to infer basal-melting rates. Kusahara et al. (2015) used a coupled ice shelf–sea ice—ocean model with a fixed LGM AIS extension, up to the continental-shelf break. In their model results, they obtained a larger basal melt value of ice shelves than PD. These large basal-melting rates occurred because the ice shelves were located at the edge of the continental-shelf break, where ice shelves are in contact with the warm CDW. However, these basal-melting values cannot be applied to the interior of the continental shelf as these waters do not penetrate so easily there. On the other hand, Obase et al. (2017) simulated basal-melting rates on an idealized PD AIS to investigate the response of basal melt rate to a changing climate. However, these basal-melting rates are not realistic and cannot be applied directly to the AIS as the grounding-line advances during the LGM affect the climatic conditions and subshelf melting. In order to investigate the impact of realistic basal-melting rates it would be necessary to account for comprehensive parameterisations or coupled ice-sheet-ocean models (Lazeroms et al., 2018; Reese et al., 2018; Favier et al., 2019; Pelle et al., 2019), which is outside of the scope of this study. Furthermore, since our and the aim was to simulate a fully advanced AIS, as suggested by geomorphological records (The RAISED Consortium, 2014), basal-melting rates were set to zero for the sake of simplicity in this work.

Another potential source of uncertainty is the employed bedrock relaxation time. A change in bedrock depth, for instance, has profound implications on the simulated AIS, as it does not only change the local sea level, but it can also facilitate (or impede) the ice advance and retreat (Philippon et al., 2006). Here we used a simple parameterisation that accounts for the elasticity of the lithosphere and a non-local response caused by lateral shift (Le Meur and Huybrechts, 1996). This formulation does not capture differences in the mantle viscosity as it applies the same spatially homogeneous time response. Nonetheless, the Antarctic bedrock is a complex component with different rheological properties. The WAIS for instance is a low-viscosity region where the bedrock deformation happens on a shorter timescale (Whitehouse, 2018; Whitehouse et al., 2019). The next generation of ice-sheet models coupled to GIA models may produce more realistic bedrock responses and hence help to improve the sea-level budget at the LGM. This can be helpful for instance to constrain the phase space of friction parameters.

## 4.4 Forcing methods

Overall, homogeneous climate anomaly-forcing relative to present day leads to a lower ice volume as a consequence of low accumulation near the ice-sheet margins (Fig. 9b). This indicates that the AIS could have stored more ice at the LGM than estimated by studies applying such a scheme. As opposed to a spatially homogeneous method, GCM outputs are capable of representing local atmospheric effects, such as atmospheric circulation changes or localized precipitation structures. Thus, recent paleo ice sheet model exercises utilise climate forcing derived from GCMs (Briggs et al., 2013; Maris et al., 2014; Sutter et al., 2019). Nevertheless, we have shown here that the spread of the simulated ice volume and ice extension for different climatic outputs can be equal to or larger than that resulting from different assumptions of basal drag. The cryosphere is a component of the Earth System that also interacts with other components, such as the atmosphere or the ocean. Therefore the configuration of the AIS (as well as other ice sheets) for the PMIP3 LGM simulations is crucial in assessing the LGM climatologies. The PMIP3 LGM simulations were forced with an AIS volume of 22.3 msle compared to PI (Abe-Ouchi et al., 2015). This ice volume largely overestimates the obtained values in this work, as well as from latest studies (Simms et al., 2019). It is clear that a significant larger AIS will create a colder and drier environment than a smaller ice sheet. Part of this effect can be partially compensated in ice-sheet models with the elevation lapse rate. Nonetheless, wind currents for instance which could affect the cloud formation and accumulation at localized regions could not be taken into account. In order to compare with PMIP3 results, the first preliminary results of PMIP4 are forced with the same AIS LGM configuration (Kageyama et al., 2020). Nonetheless, given the fact that the latest studies point to a lower ice volume, new PMIP experiments could consider the effect of a fully advanced, but smaller AIS. Another possibility is to employ fully coupled models to evaluate the LGM climatologies and the simulated LGM ice sheets.

## 4.5 Model limitations

In this study we employed a coarse resolution of 32 km. The simulation of large continental marine ice sheets has been found to be very sensitive to spatial resolution, especially at the grounding line (Pattyn et al., 2013). Grounding-line migration is a subgrid-scale process at such coarse resolutions. Ice-sheet models often use subgridding parameterisations to mimic higher resolutions at the grounding line. Nonetheless, even these parameterisations are often unable to trace the grounding-line mi-

gration correctly (Seroussi et al., 2014; Gladstone et al., 2017). Yelmo computes the fraction of grounded ice at the grounding line via a subgrid and scales the basal friction at the grounding line with the grounded ice fraction (Robinson et al., 2020). To analyze the potential implications of a higher spatial resolution, we additionally performed two LGM experiments (namely AVERAGE and COSMOS-ASO) together with the simulated PD state at 16km. We find that the simulated LGM state for a

fully advanced AIS simulates a similar volume (a difference of 0.2-0.3 msle) and has a slightly larger extension (0.2 to 0.3 million $km^2$) for both resolutions (SM, Table and Fig. S10, S11). Nonetheless, the simulated PD state is smaller for 16 km resolution than for 32 km (around 1 msle), which creates a larger LGM ice volume anomaly for 16 km. Overall, the simulated pattern and grounding-line position is similar for both resolutions (SM, Fig. S10, S11). However, it is important to mention that the equilibrated state is reached at different times for different resolution (SM, Fig. S12), pointing to the importance of

resolution for assessing grounding-line migrations.

## 5    Conclusions

The ice dynamics and the boundary climatology are two essential building blocks for the simulation of an Antarctic LGM state. Here we studied the uncertainty in LGM ice volume associated with these two factors, by investigating the effect of the representation of basal friction and of the atmospheric forcing, respectively, in simulations. First, we tested a range of potential

basal friction values of marine zones which simulated plausible LGM states. We found that for a simple linear friction law lower (larger) friction values enhance (diminish) the ice dynamics of marine zones and result in ice sheet configurations with less (more) ice volume, but still similar grounded ice extension. This led to several potential configurations of the AIS with a sea-level difference with respect to today in the range of 12.3 msle to 15.1 msle and with a total ice extension in the range of 15.7 to 15.8 million $km^2$. Then, for a particular friction configuration within the estimates of ice volume and extension,

we studied the individual sea-level contribution from simulations driven by LGM climates provided by the eleven PMIP3 participating groups. We found ice volume anomalies ranging from 9.6 to 15.4 msle and extensions of 14.6 to 15.9 million $km^2$. Our results show that the uncertainty in sea-level LGM estimates due to basal drag is similar to the uncertainty resulting from the background climatic conditions derived from PMIP3. Imposing the PMIP3 fields leads to higher precipitation rates along the Antarctic coast and hence to a larger simulated ice volume compared to using a homogeneous anomaly method.

The grounding-line advance is strongly determined by the atmospheric temperatures as well. Higher temperatures enhance ice flow reducing the ice viscosity. Because of the marine character of the WAIS, relatively high temperatures near the coast can prevent ice expansion. Thus, along with improved knowledge of basal conditions, constraining broader possible climatic changes during the LGM is imperative to be able to reduce uncertainty in the AIS volume estimates for this time period.

**Code and data availability**

Yelmo is maintained as a git repository hosted at https://github.com/palma-ice/yelmo under the licence GPL-3.0. Model documentation can be found at https://palma-ice.github.io/yelmo-docs/. The results used in this paper are archived on Zenodo (http://doi.org/10.5281/zenodo.4139169).

5 *Author contributions.* JB carried out the simulations, analyzed the results and wrote the paper. All other authors contributed to designing the simulations, analyzing the results and writing the paper.

*Competing interests.* The authors declare that they have no conflict of interest.

*Acknowledgements.* This project is TiPES contribution #23: This project has received funding from the European Union's Horizon 2020 research and innovation programme under grant agreement No 820970. This research has also been supported by the Spanish Ministry of
10 Science and Innovation project RIMA (grant agreement No CGL2017-85975-R). Alexander Robinson was funded by the Ramón y Cajal Programme of the Spanish Ministry for Science, Innovation and Universities (grant agreement No RYC-2016-20587). Simulations were performed in EOLO, the HPC of Climate Change of the International Campus of Excellence of Moncloa, funded by MECD and MICINN. We thank I. Tabone for helpful discussions. Finally, we are also grateful to the reviewers and editor for helpful comments, which led to considerable improvements of the paper.

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

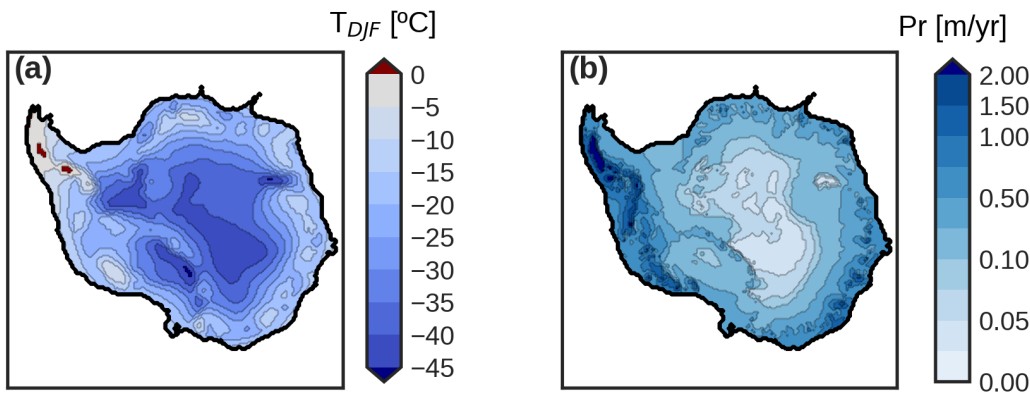

**Figure 1.** PMIP3 ensemble mean **(a)** surface summer temperature (in ºCelsius) and **(b)** annual precipitation (in m yr$^{-1}$ water equivalent) at sea level. The thick black line shows the 2000 m-depth contour.

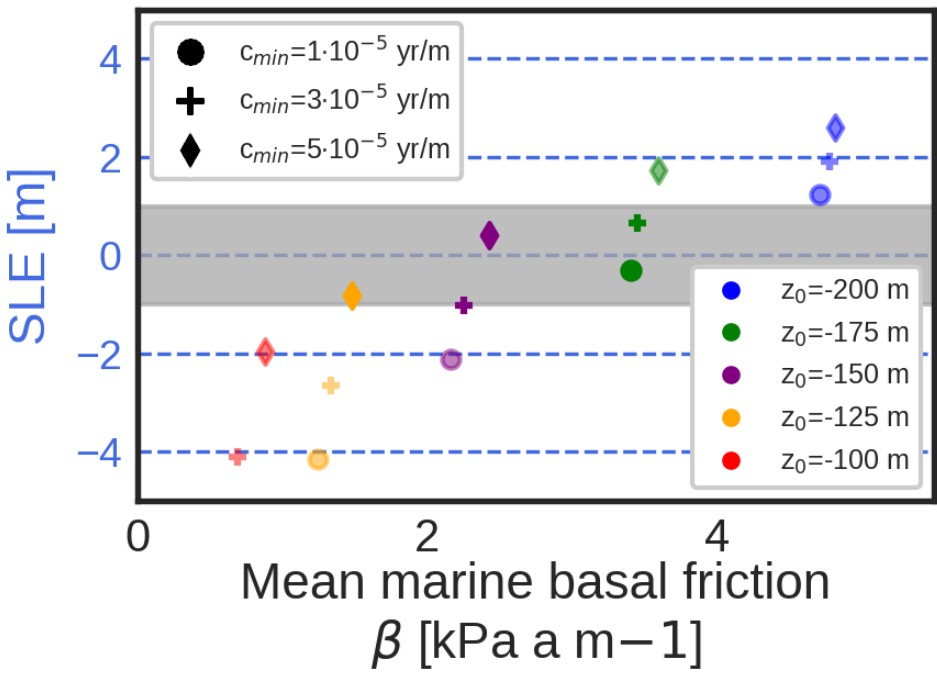

**Figure 2.** Present-day (PD) Antarctic Ice Sheet (AIS) ice volume above flotation and sea level equivalent (SLE) simulated for the explored values of friction parameters for $c_{max} = 200 \cdot 10^{-5}$ yr m$^{-1}$. The grey band represents a desviations of $\pm 1$ m from PD observations (Schaffer et al., 2016). Full colors represent simulations that fall inside the grey band.

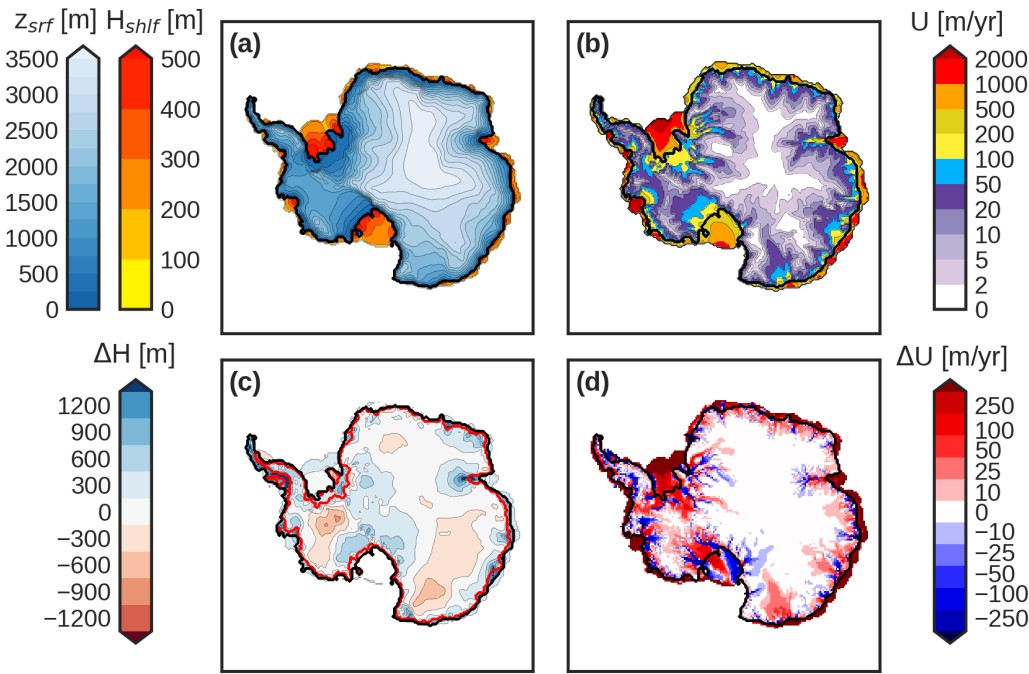

**Figure 3.** Simulated PD AIS **(a)** surface elevation (blue) and ice-shelf thickness (orange); **(b)** ice velocity; **(c)** ice thickness anomaly (simulated minus observations); **(d)** surface velocity anomaly, for the best match PD of all the ensemble mean. The thick black line corresponds to the simulated grounding-line position. The thick red line in **(c)** represents the actual grounding-line position.

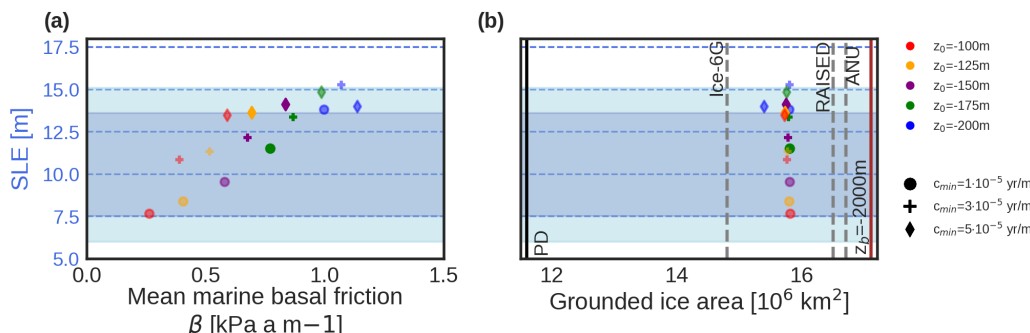

**Figure 4.** Scatter plot of the simulated LGM ice-volume anomaly (in msle, positive means ice-volume increase at the LGM) with respect to **(a)** the mean basal-drag coefficient and **(b)** the simulated grounded ice area, for the LGM simulations corresponding to different friction parameters. The dark blue horizontal area represents the SLE LGM estimates summarized by Simms et al. (2019) since 2010. The light blue area includes the uncertainties of the two extreme cases. The grey shaded vertical lines in **(b)** show the ice extension estimates from ICE-6G, The RAISED Consortium and the ANU reconstruction at the spatial resolution of our simulations (see main text). The black vertical line is the PD extension and the brown vertical line represents the computed ice area within the continental-shelf break defined as $z_b$>-2000 m. Full colors represent simulations that simulate a PD state ±1 m from PD observations.

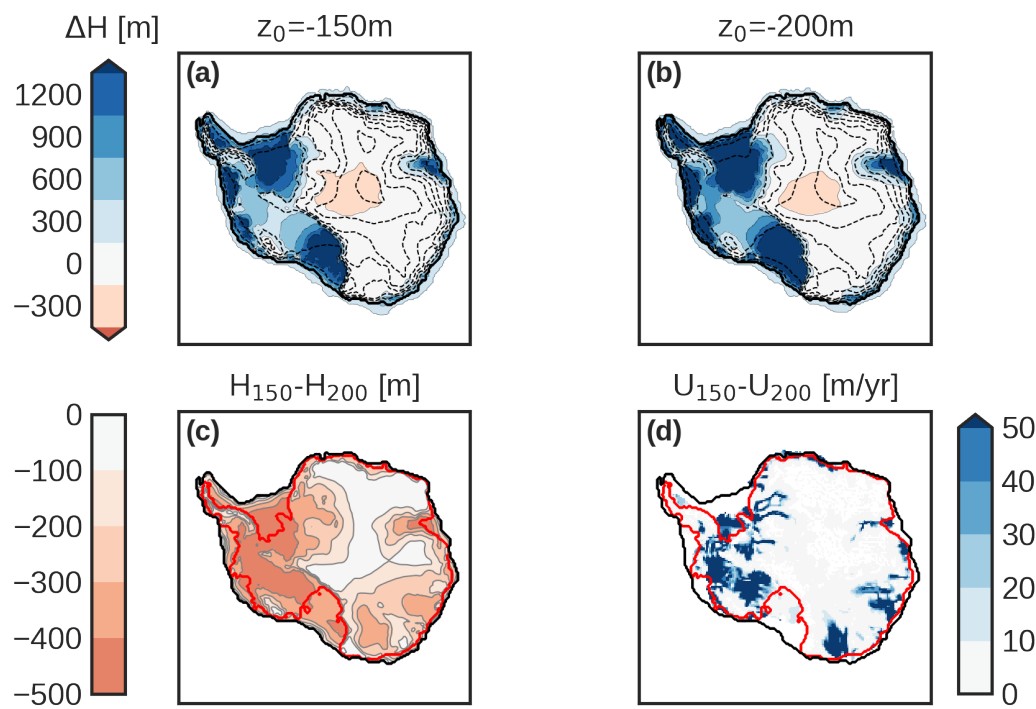

**Figure 5.** Simulated ice thickness anomaly between the simulated LGM and PD state (LGM minus PD) for $c_{\min}=1 \cdot 10^{-5}$ yr m$^{-1}$ for **(a)** $z_0$=-150 m and **(b)** $z_0$=-200 m; black discontinuous contours show surface elevation in 500 m intervals up to 3500 m above sea level. Difference in **(c)** ice thickness and **(d)** basal velocity between the two simulated LGM states (a minus b); the thick black line shows the simulated grounding-line position of $z_0$=-200 m and the thick red line the simulated PD grounding-line position.

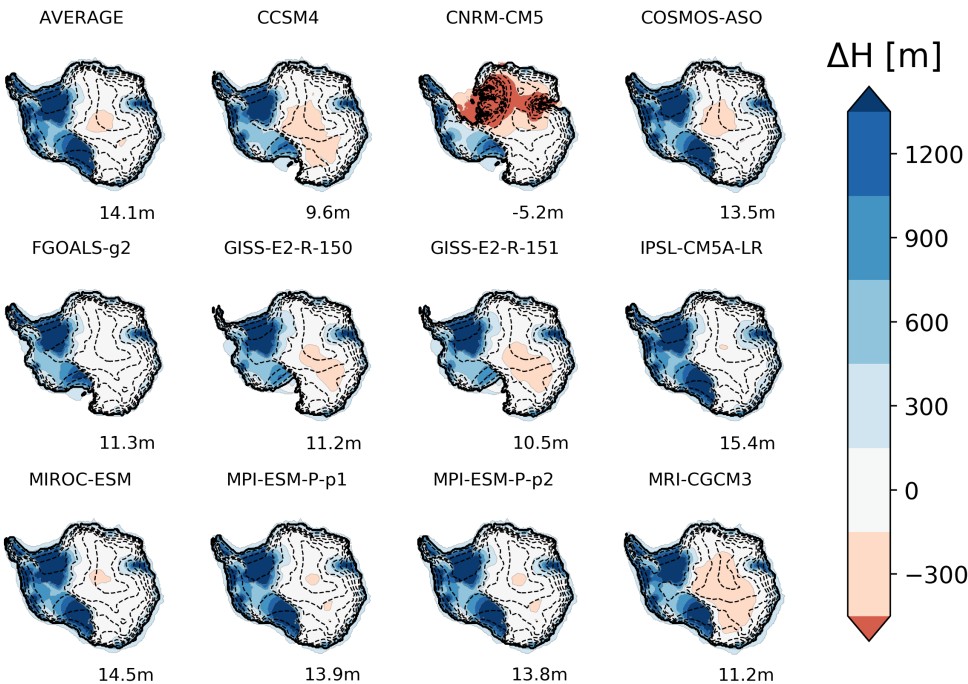

**Figure 6.** Ice thickness anomaly between the simulated LGM and PD for the PMIP3 ensemble. Black line represents the simulated LGM grounding-line position. Black discontinuous contours show surface elevation in 500 m intervals up to 3500 m. The number in each panel shows the ice volume difference between the simulated LGM and PD (LGM minus PD) in terms of msle.

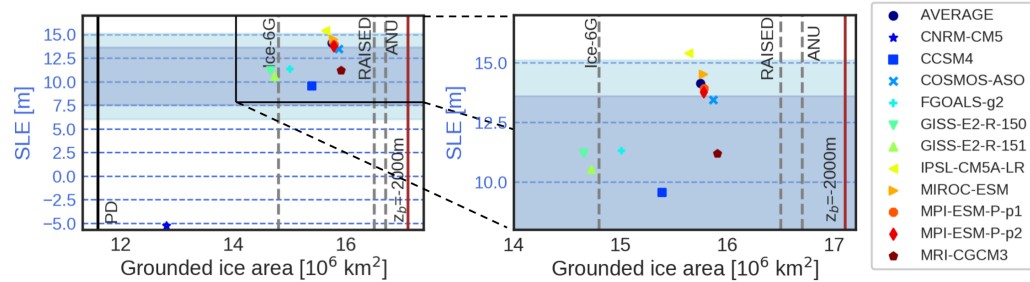

**Figure 7.** Scatter plot, as in Fig. 4, of the simulated LGM ice volume anomaly (SLE) against the grounded ice area for the PMIP3 ensemble and reference values of $z_0$=-150 m and $c_{min}$= 5·10$^{-5}$ yr m$^{-1}$.

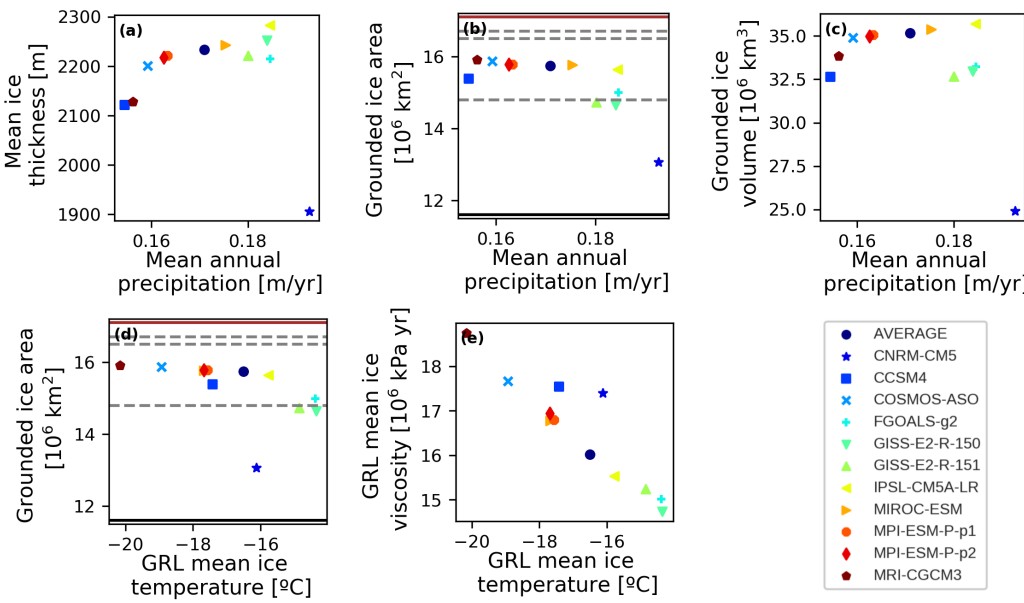

**Figure 8.** Scatter plots of **(a)** the mean ice thickness vs. the mean annual precipitation of the grounded grid points; **(b)** the grounded ice area vs. the mean annual precipitation of the grounded grid points; **(c)** the grounded ice volume vs. the mean annual precipitation of the grounded grid points; **(d)** the grounded ice area vs. the mean ice temperature at the grounding line; **(e)** the mean ice viscosity at the grounding line vs. the mean ice temperature at the grounding line. The horizontal lines in **(b)** and **(d)** represent the ice extensions described in Fig. 4.

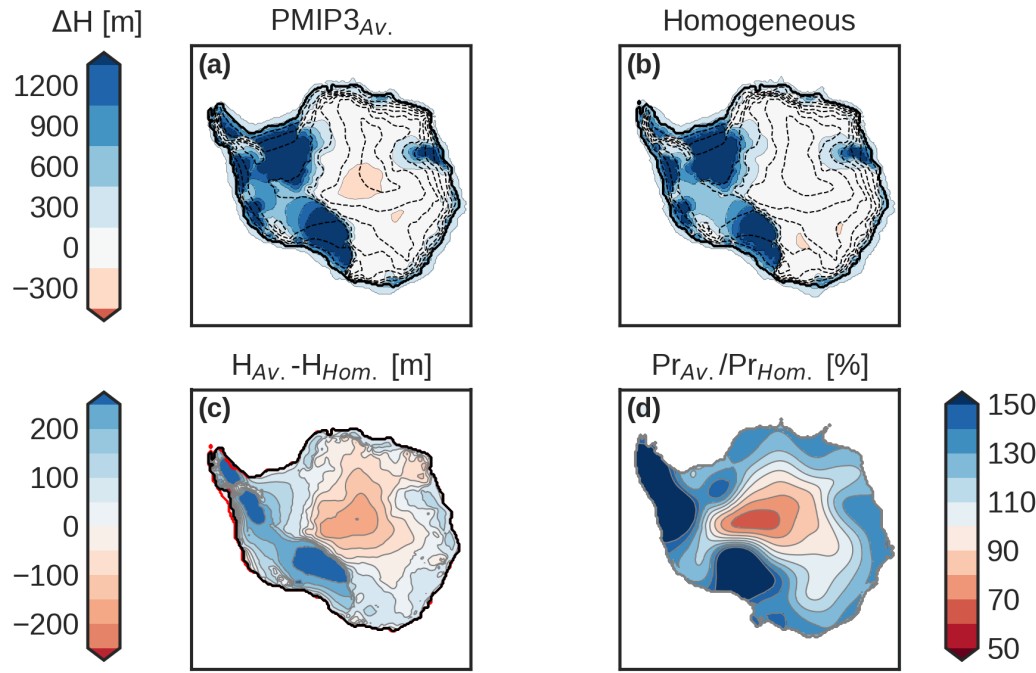

**Figure 9.** Simulated ice thickness anomaly (LGM-PD) for **(a)** the PMIP3 average snapshot and **(b)** the spatially homogeneous method with $z_0$=-150 m and $c_{min}$=5·10$^{-5}$ yrm$^{-1}$. The black discontinuous contours show surface elevation every 500 m intervals up to 3500 m above sea level. Panel **(c)** shows the ice thickness difference **(a)** minus **(b)**, where the thick red and black lines show the grounding-line position from the simulation with homogeneous and PMIP3 climatic forcing, respectively. Panel **(d)** shows the ratio of precipitation in the PMIP3 forced simulation to that of the homogeneous simulation up to the continental-shelf break ($z_b$=-2000 m).