# Peer review of "Exploring the impact of atmospheric forcing and basal drag on the Antarctic ice sheet under Last Glacial Maximum conditions"

_The Cryosphere, 2020_

## Referee Comment (RC1) · Anonymous Referee #1 · 12 Apr 2020

The paper by Blasco and others entitled "Exploring the impact of atmospheric forcing and basal boundary conditions on the simulation of the Antarctic ice sheet at the Last Glacial Maximum" presents an interesting modelling simulation of the behavior of the Antarctic Ice Sheet under different boundary conditions. I am not a numerical modeler but from my perspective the methods, approach, and analysis were well explained – I was able to understand what they were trying to accomplish. In other words the paper is well written. The authors use a suite of numerical predictions of the growth of the ice sheet in order to determine how different basal friction parameters and precipitation

and temperatures influence the predicted size of the LGM Antarctic Ice Sheet (probably not a surprise). The model setup appears to be able to replicate the present day ice sheet, building confidence that the model produces a realistic representation of the ice sheet. In addition, the size of the LGM Antarctic ice sheet is within the range of other reconstructions. Their main finding is that with lower basal friction values the ice sheet is smaller and more dynamic and the opposite is true for higher basal friction values. One important outgrowth of their work is the suggestion that most of the differences in possible ice-sheet configurations is most pronounced along the ice-sheet margins – and these locations are important for understanding the past behavior of the ice sheet (as opposed to interior sites where most of the current ice-core records are located). I think the paper provides an important contribution to our understanding of the possible past behaviors of the Antarctic Ice Sheet and will be of interest to the community. I recommend publication after minor (if any) corrections. A couple items to "chew on": 1.) The authors assume a relaxation time of 3,000 years for the GIA component (Page 5, line 8). The community is undergoing a shift in ideas on the rheology of the Earth beneath the Antarctic Ice Sheet (e.g. Whitehouse et al., 2019; Barletta et al., 2018). How sensitive is your model to this relaxation time? What happens if you use a weaker rheology? 2.) This might just be a reflection of my ignorance with models but your model is allowed to run for 80 ka (Page 6, line 12), I assume to reach some sort of equilibrium but how do we know that the ice sheet was in equilibrium. How important are the dynamics of the ice sheet leading up into the LGM for its LGM behavior? 3.) Page 3, line 3 – please give a reference for "ablation and basal melting were probably negligible at the LGM." Probably, but you could use some justification of this assumption. Other minor editorial suggestions: 1.) Page 2, line 14 – remove "up" 2.) Page 9, line 12-13: "...grounding-line from thickening, as a..." 3.) Page 9, line 14: "...viscosity such as GISS-E2-R-150..." 4.) Page 9, line 15: "...Amery Trough." 5.) Page 9, line 24: "...temperatures, which result in low viscosities. Therefore..." 6.) Page 10, line 10: "...pronounced; however, inland..." 7.) Page 10, line 30 – Please explain what you mean by "specially determinant"

---

## Referee Comment (RC2) · Johannes Sutter (Referee) · 13 Apr 2020

Blasco and colleagues examine the interesting and as of yet unresolved question of the maximum extent of the Antarctic Ice Sheet (AIS) during the Last Glacial Maximum (LGM). The existing literature on the matter exhibits a very large spread in potential LGM ice sheet configurations and volumes. Blasco et al. set out to systematically illuminate the sources of uncertainty regarding AIS LGM ice-volume and -extent reconstructions by means of paleo ice sheet modelling. This is a very valuable effort and well suited for the scope of The Cryosphere. Blasco et al. identify, and focus on, two

major sources of uncertainty when it comes to simulating the LGM state of the AIS.

1. Uncertainties in the climate forcing used to drive ice-sheet changes.

2. Uncertainties regarding the parameterisation of basal drag, which controls the efficacy of the ice drainage via outlet glaciers.

The manuscript is generally clearly structured and easy to follow. However, there are some issues which should be addressed to make this a valuable contribution to The Cryosphere.

Below I list my major concerns followed by some minor stilistic/editorial aspects.

1. The authors omit a discussion as to how their initial ice sheet configuration affects their results and conclusions:

1.1 How does the model spinup affect the final LGM extent of the AIS. Is there a thermal spinup, paleo-spinup or a "cold start". A more detailed discussion of the initial state of the ice sheet would be useful. I suggest one additional figure (this could be figure 1 or 2) which gives an overview over the initial state of the ice sheet and the present day (PD) tuning simulations (best fit, ice thickness change vs observations, ice volume and sea level equivalent, grounding line configuration, surface velocity). There are some figures in the supplement but I think an overview figure in the main manuscript is needed.

1.2 Arguably, the authors chose a relatively loose definition of a "good" present day fit with respect to sea level equivalent ice volume change (-3m to +3m). The ice volume spread at PD due to the parameterisation regime is about the same as the total LGM volume spread in their ensemble. How would the LGM spread change if more rigid conditions are applied for PD (e.g. pm 1m?). Also it seems that ice shelves are extensive in the PD tuning runs (supplementary figure 2). How does this affect grounding line advance (buttressing) as well as SMB (a very large area is gaining mass right away, whereas in reality there might have been no ice shelves).

2. The experimental setup assumes a steady state LGM-forcing for 80 ka. I understand that Blasco et al. chose an idealised setup in order to fully focus on the effects of different climatological boundary conditions and ice flow parameterisation. This is fine, however the fact should be discussed, so the reader can appreciate the potential impact on the results. In reality, full LGM-forcing was only sustained for maybe a couple of millennia, preceded by a long cooling period starting at the end of the last interglacial. The authors should include a discussion of the transient evolution of the AIS from the initial present day (PD) state to the final LGM state. How fast is equilibrium reached? Does it take several tens of thousands of years or only a couple of millennia? Is the relative homogeneity of the grounding line extent due to the long integration time under LGM conditions or the forcing? What role does the sea level boundary condition play? Actually, reading the text I was missing information whether sea level was set to LGM conditions (ca. -120 m) or PD or something in between? This is important information, as sea level alone exerts a big influence on the grounding line position via the flotation criterion. Here an additional figure would be nice which shows the transient growth of the AIS under constant LGM forcing for each ensemble member. This would elucidate the inter-ensemble differences in the pace of AIS grounding line advance and ice volume change.

3. Model resolution. This is a somewhat nasty argument as in theory very high spatial resolutions are required to adequately resolve grounding line migration. However coarse resolutions are a tried and tested instrument to allow for larger paleo ice sheet ensembles and the authors do use a sub-grid grounding line procedure to accommodate for the coarse resolution. Still, 32 km are on the rough end of currently used grid-spacing and it would be interesting the see the effect of say doubling resolution (16-km) on final LGM ice volume and extent. This does not have to be done for each and every run, but picking one single member and maybe the GCM-mean forcing would show the impact of resolution on LGM ice sheet configuration. This would mean only two additional simulations and should not take too much time.
General comments text:

The manuscript is generally well written but contains a couple of stilistic issues, redundancies, unclear sentences etc. of which I try to note a couple in the following:

Title :

I think the title is a little misleading, as you do not explicitly simulate the Last Glacial Maximum Antarctic Ice Sheet configuration per-se but rather potential equilibrium states of the AIS under LGM conditions. Below is an attempt at a slightly modified title.

Exploring the impact of atmospheric forcing and basal drag on Antarctic Ice Sheet equilibrium extent and volume under Last Glacial Maximum conditions.

Abstract: I think one interesting outcome of this study is that the ensemble spread with regard to sea level equivalent ice volume change is about the same for the tested parameterisations of basal drag as for the different GCM forcings used (both ca. 6 m). This should be mentioned in the abstract and discussion.

p3 l35: you assume a priori zero basal melt underneath ice shelves. For me this is fine, but how do you legitimize this choice? Relatively little is known about the state of CDW during the LGM, but I guess it is not to be excluded that regionally if the grounding line is located at sufficient depth, some basal melt is possible even during the LGM. Maybe a reference would be helpful here.

p4 l 31: The SMB is obtained from the difference between ice accumulation ...

p5 l10: how does this relaxation time relate to other figures used in the field? How does it affect the results?

p5 l 11 ... so an "enhancement factor" is used ...

p5 l27: This sentence is a little confusing, maybe change to : "For lower values of $z_0$, $c_b$ falls more rapidly ..."

p6 l27: how realistic is this assumption? I guess in some regions basal freeze on could be extensive and for other regions basal melt is theoretically possible. A short discussion would be helpful.

p7 l 10-12: as mentioned in my comment 1.2, how does the large SL spread in the PD simulations affect the spread at LGM. What happens if you only account for those simulations with a spread of e.g. pm 1 m.

p7 l22: suggest to change to: Here we present the simulated AIS equilibrium configuration under LGM conditions for different basal friction parameters.

p7 l22: change to: Ice volume change is converted into ...

p7 l28 : change to : ...basal friction reduces basal sliding...

p7 l29 : change to : ...also reduces ice volume...

p7 l30 : suggest to change to: We do not identify a strong impact of marine basal friction on equilibrium grounded ice area, as the final grounding line configuration is similar in all ensemble members (Fig. 2b). Comment: is this mainly due to the long integration time? How quickly is the final ice extent reached?

p8 l11 ...a slowly decreasing basal friction

p8 l26: ...a spread of 6.2 msle.

p8 l32: use other word than "appreciable", maybe "strong" ?

p9 l6 ...identify the surface temperature ...

p9 l7 maybe change to: Whereas low surface temperatures lead to similar ice extend, relatively warm surface temperature forcing results in smaller equilibrium grounding line advance.

p9 l8 change to: given the overall low surface temperature at LGM, ablation can generally be discarded as the ...

p9 l16 it is unclear here what these hypotheses are. I assume you mean:

1. The more slippery the bed the farther the ice extend and the lower the volume. 2. the colder the surface temperature the larger the ice volume 3. the higher the precip the larger the ice volume

For the reader it would be nice if the authors main hypotheses are spelled out in the beginning.

p9 l20 what is abnormal? change phrasing.

p9 l20 suggest to omit "unexpected"

p9 l22 ... the regions with grounding line advance to the continental shelf break (e.g. the Ross Basin)... Comment: how do they contribute to low ice temperature? Due to lapse rate effects? Clarify.

p9 l27 too low for what? Suggest to change to: If the viscosity is low ...

p9 l28 necessary for what?? grounding line advance?

I suggest to rephrase the whole last paragraph, beginning at "In summary". You mix climate effects on the ice sheets rheology with topographic effects due to bedrock configurations and the location of the continental shelf break under the header of "Impact of climate forcing".

The last sentence provides an important finding as it shows the impact of different SMB regimes under similar ice sheet configurations.

p9 l33. I think at the current state of art in the field it is unclear what approach is "valid" given the large persisting uncertainties in paleo ice sheet modelling (as well as ice sheet projections). I therefore suggest to rephrase to: ... is a common approach...

p10 l3. As of yet it is unclear what a "realistic" SLE is, this is something you rightly state at the beginning of the manuscript. Therefore I suggest to rephrase to : All simulations

produce SLE ice volume in the range of previously suggested figures and ice extend similar to reconstructions (e.g. Bentley et al. 2014) if using the same coefficients for basal friction and different climate forcings. Overall, consistently ...

p10 l4 change to: This is solely due to the difference in forcing, as the parameterisation of ice flow is identical.

p10 l5 change to: Since surface temperatures are not sufficient to cause surface melt, differences in ice volume and extent are exclusively due to differneces in accumulation anomalies.

p10 l7. It is evident that the main source of ice volume differences is due to changes in the WAIS configuration.

p10 suggest to change header of 4.1 to: "Role of basal friction" or similar

p10 l21 : between the end members

p10 l25 I know what you mean with "still agree with PD observations" but I suggest to to rephrase the sentence or split it in two.

p10 l26 change to: The choice of the friction law ...

p10 l30 suggest to change wording to: ...is especially relevant...

p11 l7 The simulated grounding line advance is strongly influenced by air temperature.

p11 l12 if temperatures are sufficiently cold (< 20 °C) ice full advances ...

p11 l13 The RAISED consortium shows a similar grounding line extend, albeit with two large ice shelves ... Comment: to my knowledge Bentley et al. show grounding line extend but not the presence of ice shelves but I might be mistaken? Please clarify.

p11 l29 Overall, homogenous climate anomaly-forcing relative to present day leads to a ...

p11 l32 Thus, recent paleo ice sheet model exercises utilise climate forcing derived

from GCMs

p11 l1 Nevertheless, ... resulting from different assumptions of basal drag.

p12 l4 change to: By design the modelled ice sheet could be expected to be driven towards the configuration used as a boundary condition in PMIP3. However, ...

p12 l6 ...the comparison with proxy-observations.

p12 l8 ... more accurate paleo-climate forcing will hopefully be available.

p12 l21 Imposing the PMIP3 fields, which explicitly assume an LGM ice sheet configuration, leads to higher preci...

p12 l24 I guess you mean WAIS not AIS here?

You show in your results that the uncertainties regarding basal conditions are as high as the uncertainties regarding climate forcing, this should be restated in the conclusions as I think this is an important finding of this work.

General comments figures:

Figure 3. Cmap different to read. Suggest to use simpler colormap (e.g. Red-Blue) and plot ice thickness changes relative to PD. For the surface contours I suggest using one color (e.g. gray).

Figure 4. The figure size seems overly large given that it shows less information than the following figures.

Figure 5. Tough too discern features with this color scale, I suggest something simpler (e.g. Red-Blue or similar) and plotting delta thickness with respect to present day observations instead of LGM surface elevation. This way it is easier to identify regional changes caused by the different GCM-forcings.

Figure 7. With the "jet/rainbow" color scale it is tough to discern between different ensemble members, I suggest different marker styles ("x o , ." etc) for each GCM in
addition to the colors.

Figure 8. same as Figure 5. Suggest different color scale and delta thickness instead of surface elevation. You can keep the surface contours for reference. Why is ice thickness lower in the coastal regions of the Bellinghausen Seas for the PMIP3av even though accumulation is higher? It can't be basal shelf melt as this is set to zero?

---

## Referee Comment (RC3) · Anonymous Referee #3 · 16 May 2020

Blasco et al. investigated the effects of glacial atmospheric boundary conditions and basal drag on the equilibrium state of the LGM Antarctic ice sheet (AIS). For this purpose, they performed sensitivity experiments with an ice sheet model by modifying glacial atmospheric forcing obtained from PMIP3 LGM simulations as well as parameters in the basal drag. They showed that the differences in the glacial atmospheric forcing among PMIP3 model caused discrepancies in the simulated LGM AIS by an amount of 6m SLE, which is similar to that obtained from sensitivity experiments with different basal drag. I think the content of this study matches the interest of the reader

of The Cryosphere. Furthermore, as a climate modeler, I find this result quite interesting, and think it offers valuable information to both ice sheet and climate communities. Below, I address several concerns mostly focusing on the discussion of the results.

General comments

1. I think the authors should discuss the uncertainty in the glacial atmospheric forcing arising from the ice sheet configuration used in PMIP3 LGM simulation. In the PMIP3 LGM simulations, the climate models are forced with PMIP3 LGM AIS, which has a volume of 22.3 meter SLE compared with PI (Abe-Ouchi et al. 2015). This value largely overestimates the reconstructed value of the LGM AIS (Less than + 15 m SLE), and causes an inconsistency between the LGM AIS used for climate model simulations and the simulated LGM AIS with the ice sheet model. Therefore, the author should address this problem, and suggest the climate modeling community to perform LGM simulations with a more realistic AIS, which matches the reconstruction. In addition, I have a comment on a sentence starting from P12L5 " A way to potentially test the plausibility of the employed climatic fields is to compare with ice proxies." I agree to this sentence, but again, the inconsistency in the LGM AIS used in climate models and the reconstructed LGM AIS bothers me. For example, even if some PMIP3 glacial atmospheric forcing show consistent results with available ice core data, and regarded as reasonable glacial atmospheric forcing, I don't think it is physically correct. Please add a discussion on this point in section 4.3.

2. The basal melting of the ice shelf is fixed to zero in the simulations. I think this is a reasonable simplification to focus on the main topic of this study, however you should at least discuss the potential effect of the simplification you made. For example, while Obase et al. (2017, JCLIM) show that the basal melting at the LGM largely reduced compared with PI in their simulations with regional ocean model, the basal melt of LGM was still more than 50% of the PI experiment. Based on their estimates, the simulated area and volume in your experiment can be considered as the maximum estimate, and that the uncertainties in the ice shelf basal melting can have an impact on the LGM

AIS. Please add a discussion on this topic.

3. It is interesting to see that the differences in glacial atmospheric forcing caused large discrepancies in the simulated LGM AIS, especially for that of CNRM5. While it is not the main topic of this study to understand the cause of the difference in atmospheric forcing, I think it is valuable to discuss some possible reasons. For example, the result of CNRM5 reminds me of a study by Marzocchi and Jansen (2017, GRL) who compared the sea ice among PMIP3 LGM simulation. In their Fig. 3, you can find that CNRM5 simulates the smallest austral summer sea ice extent in LGM among PMIP3 models. This will cause warmer summer temperature over the marginal region of AIS and contribute to the negative mass balance. Perhaps, you may add one or two sentences on this point.

Specific comments

P2L30-31: This sentence describes several processes, which affect the estimate of the volume of LGM AIS. However it is unclear how the modifications affect the estimate. Please add some explanations on this point. You may focus on one or two processes, which are relevant to this study.

P3L3: I mostly agree with this sentence, but is it really true that the basal meting is negligible during LGM? For example, Obase et al. (2017, JCLIM) showed with regional ocean model that there is still some basal melting occurring at LGM. Please modify this sentence in a more modest way.

P4L10: This sentence is difficult to read. Do you mean that in some models, the simulated results largely differ from ice core reconstructions? Please modify this sentence.

P6L15-20: I had difficulty understanding this sentence, since I'm not familiar with an ice sheet model. Please describe this sentence in more detail. Why do you use PD temperature field at sea level rather than surface? How does RACMO calculate the sea level temperature field? Do they assume a constant lapse rate in converting the

temperature? If so, is the value of the lapse rate identical to what you chose in your ice sheet model?

P6L19: How did you decide this value of the lapse rate? Do you have any reference for this?

P8L14: Are these results consistent with previous studies?

P9L6: How did you define ground line temperature? Does the location of grounding line depend on simulations?

P11L20-25: Please add a discussion on the role of basal melting in this sub-section.

P12L1-2: I like this finding.

P12L4-5: I don't think it's that simple. On one hand, the very thick and extensive PMIP3 LGM ice sheet can induce a drastic expansion of LGM AIS due to the large decrease in surface air temperature. However, on the other hand, the thick ice sheet will reduce the amount of precipitation, which will cause a thinning of LGM AIS, opposite to PMIP3 LGM ice sheet.

P12L8: You may cite a recent article by Kageyama et al. (2020, Climate Past Discussion), which discusses preliminary results of PMIP4 LGM experiments.

Fig. 5: It's hard to see the contour of the surface topography. Please modify this figure.

FIg.S1: I think this figure contains some important information on the reproducibly of modern Antarctic ice sheet. Please move it to the main manuscript.

References

Kageyama, M. et al.: The PMIP4-CMIP6 Last Glacial Maximum experiments: preliminary results and comparison with the PMIP3-CMIP5 simulations, Climate of the Past Discussion, https://www.clim-past-discuss.net/cp-2019-169, 2020.

Marzocchi, A., and Jansen, M. F.: Connecting Antarctic sea ice to deep-ocean circu-

lation in modern and glacial climate simulations, Geophysical Research Letters, 44, 6286-6295, 10.1002/2017gl073936, 2017.

Obase, T., Abe-Ouchi, A., Kusahara, K., Hasumi, H., and Ohgaito, R.: Responses of Basal Melting of Antarctic Ice Shelves to the Climatic Forcing of the Last Glacial Maximum and CO2 Doubling, Journal of Climate, 30, 3473-3497, 10.1175/jcli-d-15-0908.1, 2017.

---

## Editor Comment (EC1) · Olaf Eisen (Editor) · 18 May 2020

Dear Javier Blasco and coauthors,

we now received three reviews for your manuscript. As the discussion phase will be ending soon I invite you to start to prepare a revision of your manuscript. The revision should consider all the points raised by the reviewers.

After the termination of the discussion phase you will receive a formal notification from the editorial system stating that "You - as the contact author - are requested to individually respond to all referee comments (RCs) by posting final author comments on behalf of all co-authors". You can keep your response in the "final response form" very short, basically stating that you will consider all points in the revision. Only after you finished the "final response form" I can make the formal decision about the further handling of your manuscript, i.e. to invite a revision.

Best Regards, Olaf Eisen

Co-Editor in Chief

---

## Author Comment (AC1) · 27 May 2020

We thank all the referees for the thoughtful and constructive reviews that will serve to improve this work. We will respond to all points that have been arisen in the discussion in a supplementary text, together with a revised manuscript that reflects these changes.

---

## Author Response (AR1)

======== Reviewer comment 1 (anonymous) =============

I think the paper provides an important contribution to our understanding of the possible past behaviors of the Antarctic Ice Sheet and will be of interest to the community. I recommend publication after minor (if any) corrections.

We appreciate the positive review from Referee#1 and thank them for their valuable suggestions. Below you can find our response to each comment. We would like to point out that simulations have been repeated with an improved version of the Yelmo model (Robinson et al., 2020). Although specific values for each simulation have slightly changed, our main conclusions remain robust.

1.) The authors assume a relaxation time of 3,000 years for the GIA component (Page5, line 8). The community is undergoing a shift in ideas on the rheology of the Earth beneath the Antarctic Ice Sheet (e.g. Whitehouse et al., 2019; Barletta et al., 2018). How sensitive is your model to this relaxation time? What happens if you use a weaker rheology?

[Figure]

*Fig. 1 Evolution of the **(a)** grounded ice volume; **(b)** grounded ice area for different relaxation times (from 500 yr - 10000 yr) for the friction values z0 = -125 m and c_min = 5\*10^{-5} yr/m.*

[Figure]

*Fig.2: Simulated LGM AIS ice velocity for a relaxation time **(a)** of τ=500 years; **(b)** τ=10000 years after 25000 years of cold climate evolution. Dashed lines represent surface elevation contours every 500 meters up to 3500 meters. The thick black line represents the grounding-line position. **(c)** ice thickness **(d)** bedrock elevation anomaly (**(a)** minus **(b)**). The thick green/black line represents the grounding-line of **(a)/(b)**.*

Figure 1 shows the time evolution of the grounded ice volume and ice area for the reference friction parameters and the Average PMIP3 fields. All simulations yield a similar equilibrated end-state. However, not all of them reach the continental-shelf break at the same time. Figure 2 shows the simulated LGM ice sheet for a weaker rheology (a; τ=500 years) and a stronger rheology (b; τ=10000 years) after 25000 years. These results show that a weaker rheology simulates a lower ice sheet in the WAIS and especially at the Ross shelf, where it does not fully advance (Fig. 2c). Comparing the bedrock elevation differences (Fig. 2d), a weaker rheology has a more elevated bedrock at the Ross shelf, the Bellingshausen Sea and the Amundsen sea, which impedes there a complete advance.

2.) This might just be a reflection of my ignorance with models but your model is allowed to run for 80 ka (Page 6, line 12), I assume to reach some sort of equilibrium but how do we know that the ice sheet was in equilibrium. How important are the dynamics of the ice sheet leading up into the LGM for its LGM behavior?

Indeed, we ran the model for 80 ka to reach an equilibrated state. In this way we can analyze the effect of dynamics and LGM climatologies without accounting for the transient character of the ice sheet. Nonetheless, fully-LGM conditions occurred only for a couple of millennia. After the Last Interglacial (LIG; around 120 ka), global temperatures decreased slowly until they reached the LGM, at around 21 ka. Ice core records of the AIS show that temperatures were around 10 degrees colder than the PD (Jouzel et al., 2007). As temperatures became colder, the AIS advanced up to the continental-shelf break. Due to the steep slope of the continental-shelf break, the AIS is not capable of advancing further.

In order to reach an equilibrium, the total mass balance of the AIS has to be zero. Because ablation in the LGM state is most likely negligible (we argue this in the next question), only calving at the ice front and basal melt at the continental-shelf break lead to mass loss. Hence, given that during the LGM the AIS advanced to the continental-shelf break, it is very likely that accumulation rates were compensated with calving events (and potentially melting for ice shelves below the continental-shelf break, Kusahara et al., 2015), leading to an equilibrated state.

[Figure]

*Fig.3: Simulated (a) ice volume and (b) ice extent time evolution for the friction parameters ensemble. Coloured lines represent each a z0 value.*

From geomorphological records it is possible to estimate the grounding-line retreat since the LGM to the PD, but there is no ice-extent record from the LIG to the LGM. Dynamics play a crucial role in the evolution of the AIS towards its LGM state. Whereas faster dynamics facilitate a more rapid advance towards the continental-shelf break (red lines in Fig. 3b), slower dynamics need more time to reach the borders (blue lines). Nonetheless, because the simulations that reach the continental-shelf break earlier have faster ice streams, this translates into lower ice volumes (Fig. 3a).

3.)Page 3, line 3 – please give a reference for "ablation and basal melting were probably negligible at the LGM." Probably, but you could use some justification of this assumption.

Even at PD, ablation rates are almost negligible in the AIS domain except for localized regions, such as the Antarctic Peninsula (vanWessem et al., 2016, 2018). Ice core records show that the AIS was on average 10 degrees colder

than the PD at the LGM (Jouzel et al., 2007), thus, by applying a spatially homogeneous cooling, ablation rates turn to be almost negligible. Nonetheless, in this study ablation rates are computed using the output fields from the PMIP3 models. As shown in the Supplementary Material (SM), some models do show ablation in the AIS domain but we consider this very unrealistic.

Basal-melting rates on the other hand are more difficult to infer. In order to do so, it would be necessary to have a spatial map of subsurface oceanic temperatures and salinity. However, to our knowledge, there are no such paleoceanographic records for the Southern Ocean. From a modelling perspective, PMIP3 fields from the LGM also give the simulated outputs of salinity and oceanic surface temperature. However, these models use an AIS LGM state up to the continental-shelf break (Abe-Ouchi et al., 2015) and hence it is not valid for computing at the interior of the continental-shelf. Therefore, because including basal-melting rates would add a degree of difficulty and the aim was to simulate a fully advanced AIS, basal-melting rates were set to zero for the sake of simplicity in this work.

We rephrased the above sentence in the manuscript for:
"Ablation rates at the PD are almost zero except for localized areas (van Wessem et al., 2016, 2018). Because the LGM is a colder period, around 10 degrees as shown by ice core records (Jouzel et al., 2007), ablation rates in the LGM would have been probably negligible. On the other hand, basal melting rates from the LGM are difficult to estimate due to the scarcity of oceanic temperature reconstructions. Nonetheless, geomorphological records point to a fully advanced AIS during the LGM (The Raised Consortium, 2014). This could hint to low basal-melting rates inside the continental-shelf break."

Other minor editorial suggestions:
1.) Page 2, line 14: remove "up"
Done.

2.) Page 9,line 12-13: "...grounding-line from thickening, as a..."
Done.

3.) Page 9, line 14: "...viscosity such as GISS-E2-R-150..."
Done.

4.) Page 9, line 15: "...Amery Trough."
Done.

5.) Page 9, line 24: "...temperatures, which result in low viscosities. Therefore..."
Done.

6.) Page 10, line 10: "...pronounced; however, inland..."
Done.

7.) Page 10, line 30 – Please explain what you mean by "specially determinant"
Because this statement can cause confusion we changed the sentence

"However, the importance of saturated tills is specially determinant for transient simulations with a retreating grounding line."

to:

"However, the aim of this work was to study the uncertainty associated with the basal drag parameters, rather than assessing the uncertainty for different friction laws."

======== Reviewer comment 2 (Johannes Sutter) =============

The manuscript is generally clearly structured and easy to follow. However, there are some issues which should be addressed to make this a valuable contribution to The Cryosphere. Below I list my major concerns followed by some minor stilistic/editorial aspects.

We are grateful to Johannes Sutter for bringing up several key points that will serve to improve the manuscript. We have addressed these concerns below. We would like to point out that simulations have been repeated with an improved version of the Yelmo model (Robinson et al., 2020). Although specific values for each simulation have slightly changed, our main conclusions are robust.

1. The authors omit a discussion as to how their initial ice sheet configuration affects their results and conclusions:
1.1 How does the model spin up affect the final LGM extent of the AIS. Is there a thermal spin up, paleo-spin up or a "cold start". A more detailed discussion of the initial state of the ice sheet would be useful. I suggest one additional figure (this could be figure 1 or 2) which gives an overview over the initial state of the ice sheet and the present day (PD) tuning simulations (best fit, ice thickness change vs observations, ice volume and sea level equivalent, grounding line configuration, surface velocity). There are some figures in the supplement but I think an overview figure in the main manuscript is needed.

The LGM and the PD simulations start from the same initial state, mainly the PD topographic variables (bedrock, ice thickness, masks, etc.). The remaining variables, namely dynamics and thermodynamics, are derived from boundary conditions. Then LGM and PD conditions are run for 80 kyr under the respective constant climatic conditions, hence a "cold" start for the LGM.

We added a figure of the best simulated PD (which from now on is the new reference state in the manuscript, Fig. 1). A discussion can be found in the next point.

[Figure]

Figure 1: Simulated PD AIS *(a)* surface elevation (blue) and ice-shelf thickness (orange); *(b)* ice velocity; *(c)* ice thickness anomaly (simulated minus observations); *(d)* surface velocity anomaly, for the best match PD of all the ensemble mean. The thick black line corresponds to the simulated grounding-line position. The thick red line in *(c)* represents the actual grounding-line position.

1.2 Arguably, the authors chose a relatively loose definition of a "good" present day fit with respect to sea level equivalent ice volume change (-3m to +3m). The ice volume spread at PD due to the parameterisation regime is about the same as the total LGM volume spread in their ensemble. How would the LGM spread change if more rigid conditions are applied for PD (e.g. pm 1m?). Also it seems that ice shelves are extensive in the PD tuning runs (supplementary figure 2). How does this affect grounding line advance (buttressing) as well as SMB (a very large area is gaining mass right away, whereas in reality there might have been no ice shelves).

The extension of the PD ice shelves was improved with the new version of Yelmo and is in better agreement with observations (Fig. 1 and new SM figures). If we apply more rigid conditions, such as ±1 m, then the spread reduces to 3.8 m (from 10.3 to 14.1 msle). We now only focus on that range rather than ±3 m in the new manuscript version.

We added in the Discussion section:

"The simulated PD configurations show a slightly more advanced grounding line in the WAIS compared to the observations, especially at the Ronne shelf. Also the ice thickness in the interior of the WAIS is systematically lower than observations. Both features can be partially explained by the basal-drag parameterisation used. Our parameterisation enhances sliding for deeper bedrock. The WAIS is in its vast majority a marine ice sheet, where bedrock depths can reach up to 2000 m in the interior regions. Thus we systematically simulate a lower WAIS, as we overestimate the ice flow at the interior. This, in addition, promotes the grounding line to advance. Nonetheless, this parameterisation allows for a precise tracing of ice streams. Except in the Larsen embayment, ice shelves generally show a slightly larger extension than observations. Because larger ice shelves allow for more ice accumulation and exert a backward force, it also helps the grounding-line to advance. Thus, the more advanced grounding line in the Ronne, Amundsen sea and Amery shelves could be additionally explained by the backward force exerted by ice shelves. Nonetheless, the overall picture of the simulated AIS fits well with observations in terms of grounding-line position as well as simulated ice volumes."

2. The experimental setup assumes a steady state LGM-forcing for 80 ka. I understand that Blasco et al. chose an idealised setup in order to fully focus on the effects of different climatological boundary conditions and ice flow parameterisation. This is fine,however the fact should be discussed, so the reader can appreciate the potential impact on the results. In reality, full LGM-forcing was only sustained for maybe a couple of millennia, preceded by a long cooling period starting at the end of the last interglacial.The authors should include a discussion of the transient evolution of the AIS from the initial present day (PD) state to the final LGM state. How fast is equilibrium reached?Does it take several tens of thousands of years or only a couple of millennia? Is the relative homogeneity of the grounding line extent due to the long integration time under LGM conditions or the forcing? What role does the sea level boundary condition play?Actually, reading the text I was missing information whether sea level was set to LGM conditions (ca. -120 m) or PD or something in between? This is important information,as sea level alone

exerts a big influence on the grounding line position via the flotation criterion. Here an additional figure would be nice which shows the transient growth of the AIS under constant LGM forcing for each ensemble member. This would elucidate the inter-ensemble differences in the pace of AIS grounding line advance and ice volume change.

Indeed, we left out information about the boundary sea-level stand. In the simulations, it is set to -120 m. This has been added to the Methods section.

Yes, the large integration time contributes to a similar extension for all the simulations. Nonetheless, this extension is reached in all simulations at most after 45kyr. Assuming that the LGP occurred for almost 100 kyr we think that it is realistic that all the ice sheets fully expanded at the LGM. We added in the Discussion section:

"In this study we assumed steady-state LGM and PD conditions to investigate the effect of climatological boundary conditions and basal drag parameterisation. Of course, this represents a simplification of reality, as full LGM conditions only occurred for a couple of millennia. In a transient simulation, the results would additionally include a potential internal drift, which we tried to avoid. Although simulations were forced during 80 kyr under steady LGM conditions, equilibrated states were reached after only 30 to 40 kyr (see SM). Given that the LGP was a cold and sufficiently long period in the Antarctic domain, constant LGM conditions should be enough to stabilize the AIS near its real LGM state."

We also added to the SM the transient evolution of the whole ensemble (Figures 2, 3).

[Figure]

Figure 2: Simulated *(a)* ice volume and *(b)* ice extent time evolution for the friction parameters ensemble. Coloured lines represent each a $z_0$ value.

[Figure]

Figure 3: Simulated *(a)* ice volume and *(b)* ice extent time evolution for the whole PMIP3 ensemble and the reference friction parameters $z_0 = -125m$ and $c_{min} = 5*10^{-5}$ yr/m.

3. Model resolution. This is a somewhat nasty argument as in theory very high spatial resolutions are required to adequately resolve grounding line migration. However coarse resolutions are a tried and tested instrument to allow for larger paleo ice sheet ensembles and the authors do use a sub-grid grounding line procedure to accommodate for the coarse resolution. Still, 32 km are on the rough end of currently used grid-spacing and it would be interesting to see the effect of say doubling resolution(16-km) on final LGM ice volume and extent. This does not have to be done for each and every run, but picking one single member and maybe the GCM-mean forcing would show the impact of resolution on LGM ice sheet configuration. This would mean only two additional simulations and should not take too much time.

We have followed the reviewer's suggestion. Here we show the results in terms of sea-level equivalent (SLE) for 32km and 16km for two PMIP3 members: the COSMOS-ASO as well as the mean forcing of the whole ensemble (AVERAGE). In addition, simulations for PD forcing based on observations were carried out for both spatial resolutions.

[Figure]

*Figure 4: Ice volume evolution in terms of SLE and grounded ice area evolution for two PMIP3 members and the average with 32km and 16km resolution.*

|  | V 32km [msle] | V 16km [msle] | A 32km [$10^6$ km$^2$] | A 16km [$10^6$ km$^2$] |
|---|---|---|---|---|
| AVERAGE | 72.8 | 73.0 | 15.7 | 16.0 |
| COSMOS-ASO | 72.2 | 72.3 | 15.9 | 16.1 |
| PD | 58.7 | 57.6 | 12.9 | 13.0 |

*Table summarizing the simulated ice volume (in msle) and grounded ice extension (in $10^6$ km$^2$) for different resolutions.*

The simulated LGM state for AVERAGE 16km and COSMOS-ASO 16km has a similar ice volume than for 32km resolution. However, the simulated PD state is smaller for 16km resolution than for 32km (around 1 msle), which creates a larger LGM ice volume anomaly as it is measured with respect to the simulated PD state. The simulated LGM state for 16km is more extended (0.3 and 0.2 million km² respectively) which allows for more ice accumulation, and results in a slightly larger LGM ice volume per se.

Overall, the simulated LGM snapshots are similar for both resolutions, with a similar ice thickness anomaly pattern (Figure 5, 6).

[Figure]

*Figure 5: Upper row: Simulated surface elevation and ice shelf thickness for AVERAGE, COSMOS-ASO and PD at 32km resolution. Lower row: ice thickness anomaly with respect to the simulated PD (LGM-PD). In the case of the PD the ice thickness anomaly is drawn with respect to observations (simulated PD - observed PD).*

[Figure]

*Figure 6: As Figure 5 but for a horizontal resolution of 16 km.*

We added these Figures to the SM and added this paragraph in the discussion section:

"**Model limitations**

In this study we employed a coarse resolution of 32km. The simulation of large continental marine ice sheets has been found to be very sensitive to spatial resolution, especially at the grounding line (Pattyn et al., 2012). Grounding-line migration is a subgrid-scale process at such coarse resolutions. Ice-sheet models often use subgridding parameterisations to mimic higher resolutions at the grounding line. Nonetheless, even these parameterisations are often unable to trace the grounding-line migration correctly (Seroussi et al, 2014; Gladstone et al., 2017). Yelmo computes the fraction of grounded ice at the grounding line via subgrid and scales the basal friction at the grounding line with the grounded ice fraction (Robinson et al., 2019). To analyze the potential implications of a higher spatial resolution, we additionally performed two LGM experiments (namely AVERAGE and COSMOS-ASO) together with the simulated PD state at 16km.

We find that the simulated LGM state  for a fully advanced AIS simulates a similar volume (a difference of 0.2-0.3 msle) and has a slightly larger extension (0.2 to 0.3 million km$^2$) for both resolutions (SM). Nonetheless, the simulated PD state is smaller for 16km resolution than for 32km (around 1 msle), which creates a larger LGM ice volume anomaly for 16km. Overall, the simulated pattern and grounding-line position is similar for both resolutions (SM). However, it is important to mention that the equilibrated state is reached at different times for different resolution (SM), pointing to the importance of resolution for assessing grounding-line migrations."

General comments text: The manuscript is generally well written but contains a couple of stilistic issues, redundancies, unclear sentences etc. of which I try to note a couple in the following:

Title: I think the title is a little misleading, as you do not explicitly simulate the Last Glacial Maximum Antarctic Ice Sheet configuration per-se but rather potential equilibrium states of the AIS under LGM conditions. Below is an attempt at a slightly modified title. Exploring the impact of atmospheric forcing and basal drag on Antarctic Ice Sheet equilibrium extent and volume under Last Glacial Maximum conditions.

We appreciate the suggestion, and changed the title from
"Exploring the impact of atmospheric forcing and basal boundary conditions on the simulation of the Antarctic ice sheet at the Last Glacial Maximum"
to
"Exploring the impact of atmospheric forcing and basal drag on the Antarctic ice sheet under Last Glacial Maximum conditions"

Abstract: I think one interesting outcome of this study is that the ensemble spread with regard to sea level equivalent ice volume change is about the same for the tested parameterisations of basal drag as for the different GCM forcings used (both ca. 6 m).This should be mentioned in the abstract and discussion.
We added in the Abstract:
"Overall, we find that the spread in the simulated ice volume for the tested basal drag parameterisations is about the same range as for the different GCM forcings (4 to 5 m)."

p3 l35: you assume a priori zero basal melt underneath ice shelves. For me this is fine,but how do you legitimize this choice? Relatively little is known about the state of CDW during the LGM, but I guess it is not to be excluded that regionally if the grounding line is located at sufficient depth, some basal melt is possible even during the LGM. Maybea reference would be helpful here.

We added the sentence:
"Ablation rates at the PD are almost zero except for localized areas (van Wessem et al., 2016, 2018). Because the LGM is a colder period, around 10 degrees as shown by ice core records (Jouzel et al., 2007), ablation rates in the LGM would have been probably negligible. On the other hand, basal melting rates from the LGM are difficult to estimate due to the scarcity of oceanic temperature reconstructions. Nonetheless, geomorphological records point to a fully advanced AIS during the LGM (The Raised Consortium, 2014). This could hint to low basal-melting rates inside the continental-shelf break."

We also added two paragraphs in the Discussion section saying:

"In this study, no basal melting was considered during the LGM. Of course, this is a vast simplification of reality. Unfortunately, reconstructions of ocean subsurface temperatures at the LGM are not available, so that the geological evidence for basal melt is lacking. As shown in Golledge et al., (2012), oceanic forcing leads to a dynamic response of LGM ice streams in the WAIS. If basal melt would have been considered, this would have most likely reduced the total LGM ice volume and affected its extension. Thus, our results represent an upper limit which would reduce when oceanic forcing is considered.

From the point of view of modelling, there have been some attempts to infer basal-melting rates. Kusahara et al., (2015) used a coupled ice-shelf-sea-ice-ocean model with a fixed LGM AIS extension, up to the continental-shelf break. In their model results, they obtained a larger basal melt value of ice shelves than PD. These large basal-melting rates occurred because the ice shelves were located at the edge of the continental-shelf

break, where ice shelves are in contact with the warm CDW. However, these basal-melting values cannot be applied to the interior of the continental shelf as these waters do not penetrate so easily there. On the other hand, Obase et al., (2017) simulated basal-melting rates on an idealized PD AIS to investigate the response of basal melt rate to a changing climate. However, these basal-melting rates are not realistic and cannot be applied directly to the AIS as the grounding-line advances during the LGM affect the climatic conditions and subshelf melting. In order to investigate the impact of realistic basal-melting rates it would be necessary to account for comprehensive parameterisations or coupled ice-sheet-ocean models (Lazeroms et al., 2018; Reese et al., 2018; Favier et al., 2019; Pelle et al., 2020), which is outside of the scope of this study. Furthermore, since our aim was to simulate a fully advanced AIS, as suggested by geomorphological records (The Raised Consortium, 2014), basal-melting rates were set to zero for the sake of simplicity in this work."

p4 l 31: The SMB is obtained from the difference between ice accumulation …
Done

p5 l10: how does this relaxation time relate to other figures used in the field? How does it affect the results?

[Figure]

*Fig. 7 Evolution of the **(a)** grounded ice volume; **(b)** grounded ice area for different relaxation times (from 500 yr - 10000 yr) for the friction values $z_0 = -125$ m and $c_{min} = 5*10^{-5}$ yr/m.*

[Figure]

*Fig.8: Simulated LGM AIS ice velocity for a relaxation time **(a)** of τ=500 years; **(b)** τ=10000 years after 25000 years of cold climate evolution. Dashed lines represent surface elevation contours every 500 meters up to 3500 meters. The thick black line represents the grounding-line position. **(c)** ice thickness **(d)** bedrock elevation anomaly (**(a)** minus **(b)**). The thick green/black line represents the grounding-line of **(a)**/**(b)**.*

Figure 7 shows the time evolution of the grounded ice volume and ice area for the reference friction parameters and the Average PMIP3 fields. All simulations yield a similar equilibrated end-state. However, not all of them reach the continental-shelf break at the same time. Figure 2 shows the simulated LGM ice sheet for a weaker rheology (a; τ=500 years) and a stronger rheology (b; τ=10000 years) after 25000 years. These results show that a weaker rheology simulates a lower ice sheet in the WAIS and especially at the Ross shelf, where it does not fully advance (Fig. 2c). Comparing the bedrock elevation differences (Fig. 2d), a weaker rheology has a more elevated bedrock at the Ross shelf, the Bellingshausen Sea and the Amundsen sea, which impedes there a complete advance.

p5 l 11 ... so an "enhancement factor" is used …
Done

p5 l27: This sentence is a little confusing, maybe change to : "For lower values of $z_0$,$c_b$ falls more rapidly ..."
Done

p6 l27: how realistic is this assumption? I guess in some regions basal freeze on could be extensive and for other regions basal melt is theoretically possible. A short discussion would be helpful.
Discussion added (see above).

p7 l 10-12: as mentioned in my comment 1.2, how does the large SL spread in the PD simulations affect the spread at LGM. What happens if you only account for those simulations with a spread of e.g. pm 1 m.
The new manuscript version only accounts for simulations with a spread of ±1 m.

p7 l22: suggest to change to: Here we present the simulated AIS equilibrium configuration under LGM conditions for different basal friction parameters.
Done

p7 l22: change to: Ice volume change is converted into …
Done

p7 l28 : change to : ...basal friction reduces basal sliding…
Done

p7 l29 : change to : ...also reduces ice volume…
Done

p7 l30 : suggest to change to: We do not identify a strong impact of marine basal friction on equilibrium grounded ice area, as the final grounding line configuration is similar in all ensemble members (Fig. 2b). Comment: is this mainly due to the long integration time? How quickly is the final ice extent reached?

Done. As mentioned above, this is partly due to the long integration time, however, assuming that the LGP occurred for almost 100kyr we think that it is realistic that all the simulated ice sheets  fully expand at the LGM.

p8 l11 ...a slowly decreasing basal friction
Done

p8 l26: ...a spread of 6.2 msle.
Done

p8 l32: use other word than "appreciable", maybe "strong" ?
Done

p9 l6 ...identify the surface temperature …
Done

p9 l7 maybe change to: Whereas low surface temperatures lead to similar ice extend,relatively warm surface temperature forcing results in smaller equilibrium grounding line advance.
Done

p9 l8 change to: given the overall low surface temperature at LGM, ablation can generally be discarded as the …
Done

p9 l16 it is unclear here what these hypotheses are. I assume you mean:1. The more slippery the bed the farther the ice extend and the lower the volume. 2. the colder the surface temperature the larger the ice volume 3. the higher the precip the larger the ice volume. For the reader it would be nice if the authors main hypotheses are spelled out in the beginning.

We changed the paragraph to:

"The CNRM-CM5 model simulates the smallest AIS LGM for all the PMIP3 models. This model expands partly at the Ross shelf and Antarctic Peninsula zone, but collapses completely in the Ronne and Amery shelf, leading to ice free zones in the EAIS and a lower ice volume than the PD (Fig. 5). This occurs due to the presence of ablation in these regions (see SI, Fig. S8). Such a configuration is highly unlikely compared with sea-level and ice extension reconstructions from the LGM. We will discuss later possible explanations for this behaviour.

In summary, we find that the choice of the boundary climate conditions is crucial for the simulated LGM ice sheet. On one hand, the atmospheric temperatures near the coastal regions control the ice extension through viscosity. If the viscosity is low, then the ice flows too fast, preventing the necessary thickening for advancing towards the continental-shelf break. Particularly, if the bedrock is too deep, the ice sheet's expansion will be hampered. Secondly, if the ice sheet extends close to the continental-shelf break, then the accumulation pattern will determine the total amount of ice volume. We find that for fully extended ice sheets (IPSL-CM5A-LR and MRI-CGCM3), the sea-level difference due to accumulation differences is about 4.2 msle."

Done

Done

This part was changed in the new manuscript, however, the lower ice temperatures occur partly due to lapse rate effects but also to the employed PMIP3 field, which can have warmer temperatures at the coastal regions.

Done

The new paragraph reads now:
"In summary, we find that the choice of the boundary climate conditions is crucial for the simulated LGM ice sheet. Atmospheric temperatures have a direct impact on the ice flow of ice sheets. Warmer temperatures lead to lower ice viscosities, enhancing ice flow. A faster flow leads to thinner ice.

On one hand, the atmospheric temperatures near the coastal regions control the ice extension through viscosity. If the viscosity is too low, then the ice flows too fast, preventing the necessary thickening. Particularly, if the bedrock is too deep, the ice sheet's expansion will be hampered. Secondly, if the ice sheet extends close to the continental-shelf break, then the accumulation pattern will determine the total amount of ice volume. We find

that for similarly extended ice sheets (IPSL-CM5A-LR and MRI-CGCM3), the sea-level difference due to accumulation differences is about 3.5 msle"

p9 l33. I think at the current state of art in the field it is unclear what approach is "valid"given the large persisting uncertainties in paleo ice sheet modelling (as well as ice sheet projections). I therefore suggest to rephrase to: ... is a common approach…
Done

p10 l3. As of yet it is unclear what a "realistic" SLE is, this is something you rightly state at the beginning of the manuscript. Therefore I suggest to rephrase to : All simulations produce SLE ice volume in the range of previously suggested figures and ice extend similar to reconstructions (e.g. Bentley et al. 2014) if using the same coefficients for basal friction and different climate forcings. Overall, consistently …
Done

p10 l4 change to: This is solely due to the difference in forcing, as the parameterisation of ice flow is identical.
Done

p10 l5 change to: Since surface temperatures are not sufficient to cause surface melt,differences in ice volume and extent are exclusively due to differences in accumulation anomalies.
Done

p10 l7. It is evident that the main source of ice volume differences is due to changes in the WAIS configuration.
Done

p10 suggest to change header of 4.1 to: "Role of basal friction" or similar
Done

p10 l21 : between the end members
Done

p10 l25 I know what you mean with "still agree with PD observations" but I suggest to rephrase the sentence or split it in two.

We changed the paragraph to "The dynamical state of the LGM remains a source of uncertainty as there are no observations from that time period of the AIS configuration. To study potentially possible AIS LGM dynamical states, we covered a range of friction values which lead to realistic LGM and PD configurations. [...] For example, an AIS that extends up to the continental-shelf break, but with a relatively low volume increase, can be achieved through a very dynamically active ice sheet. In that case, marine-based regions, and more specifically the WAIS, have the potential to maintain fast ice streams at the LGM."

p10 l26 change to: The choice of the friction law …

Done

p10 l30 suggest to change wording to: ...is especially relevant…

To avoid the confusion pointed out by one of the reviewers, we changed the sentence to:

"However, the aim of this work was to study the uncertainty associated with the bedrock friction parameter, rather than assessing the uncertainty for different friction laws."

p11 l7 The simulated grounding line advance is strongly influenced by air temperature.

Done

p11 l12 if temperatures are sufficiently cold ($< 20 \circ$C) ice full advances …

Done

p11 l13 The RAISED consortium shows a similar grounding line extend, albeit with two large ice shelves ... Comment: to my knowledge Bentley et al. show grounding line extend but not the presence of ice shelves but I might be mistaken? Please clarify.

Indeed, they show grounding-line extent but no ice shelves. We changed the sentence to:

"The RAISED Consortium has a similar extension, but presents retreated areas at the margins of the Ronne shelf, which we are not able to simulate."

p11 l29 Overall, homogenous climate anomaly-forcing relative to present day leads to a …
Done

p11 l32 Thus, recent paleo ice sheet model exercises utilise climate forcing derived from GCMs
Done

p11 l1 Nevertheless, ... resulting from different assumptions of basal drag.
Done

p12 l4 change to: By design the modelled ice sheet could be expected to be driven towards the configuration used as a boundary condition in PMIP3. However, …
p12 l6 ...the comparison with proxy-observations.
p12 l8 ... more accurate paleo-climate forcing will hopefully be available.
This last part has changed as suggested by another reviewer.

p12 l21 Imposing the PMIP3 fields, which explicitly assume an LGM ice sheet configuration, leads to higher preci…
We deleted "whose climate simulations include dynamic adjustment to the LGM boundary conditions," as this is not that simple. As pointed out by another reviewer "On one hand, the very thick and extensive PMIP3 LGM ice sheet can induce a drastic expansion of LGM AIS due to the large decrease in surface air temperature. However, on the other hand, the thick ice sheet will reduce the amount of precipitation, which will cause a thinning of LGM AIS, opposite to PMIP3 LGM ice sheet"

p12 l24 I guess you mean WAIS not AIS here? You show in your results that the uncertainties regarding basal conditions are as high as the uncertainties regarding climate forcing, this should be restated in the conclusions as I think this is an important finding of this work.
Yes, we meant WAIS there. We added the sentence "Our results show that the uncertainty in sea-level LGM estimates due to basal drag is similar to the uncertainty resulting from the background climatic conditions derived from PMIP3.".

General comments figures:

Figure 3. Cmap different to read. Suggest to use simpler colormap (e.g. Red-Blue)and plot ice thickness changes relative to PD. For the surface contours I suggest using one color (e.g. gray).

Done. Surface contours were changed to discontinuous black lines.

Figure 4. The figure size seems overly large given that it shows less information than the following figures.

This figure was moved to SM.

Figure 5. Tough too discern features with this color scale, I suggest something simpler (e.g. Red-Blue or similar) and plotting delta thickness with respect to present day observations instead of LGM surface elevation. This way it is easier to identify regional changes caused by the different GCM-forcings.

Done

Figure 7. With the "jet/rainbow" color scale it is tough to discern between different ensemble members, I suggest different marker styles ("x o , ." etc) for each GCM in addition to the colors.

Done (also for Figure 6)

Figure 8. same as Figure 5. Suggest different color scale and delta thickness instead of surface elevation. You can keep the surface contours for reference. Why is ice thickness lower in the coastal regions of the Bellinghausen Seas for the PMIP3av even though accumulation is higher? It can't be basal shelf melt as this is set to zero?

Done. These results have slightly changed with the new Yelmo version.

======== Reviewer comment 3 (anonymous) =============

I think the content of this study matches the interest of the reader of The Cryosphere. Furthermore, as a climate modeler, I find this result quite interesting, and think it offers valuable information to both ice sheet and climate communities. Below, I address several concerns mostly focusing on the discussion of the results.

We thank the thoughtful and constructive review from Referee#3. Below you can find our response to each comment. We would like to point out that simulations have been repeated with an improved version of the Yelmo model (Robinson et al., 2020). Although specific values for each simulation have slightly changed, our main conclusions remain robust.

General comments:
1. I think the authors should discuss the uncertainty in the glacial atmospheric forcing arising from the ice sheet configuration used in PMIP3 LGM simulation. In the PMIP3 LGM simulations, the climate models are forced with PMIP3 LGM AIS, which has a volume of 22.3 meter SLE compared with PI (Abe-Ouchi et al. 2015). This value largely overestimates the reconstructed value of the LGM AIS (Less than + 15 m SLE), and causes an inconsistency between the LGM AIS used for climate model simulations and the simulated LGM AIS with the ice sheet model. Therefore, the author should address this problem, and suggest the climate modeling community to perform LGM simulations with a more realistic AIS, which matches the reconstruction. In addition,I have a comment on a sentence starting from P12L5 " A way to potentially test the plausibility of the employed climatic fields is to compare with ice proxies." I agree to this sentence, but again, the inconsistency in the LGM AIS used in climate models and the reconstructed LGM AIS bothers me. For example, even if some PMIP3 glacial atmospheric forcing show consistent results with available ice core data, and regarded as reasonable glacial atmospheric forcing, I don't think it is physically correct. Please add a discussion on this point in section 4.3.

Indeed, the employed LGM AIS is clearly larger, not only than the simulated in this work, but also in comparison with other recent studies (Simms et al., 2019).

We added a paragraph in the Discussion section:

"The cryosphere is a component of the Earth System that also interacts with other components, such as the atmosphere or the ocean. Therefore the configuration of the AIS (as well as other ice sheets) for the PMIP3 LGM simulations plays a crucial role in LGM climatologies. We note that the PMIP3 LGM simulations were forced with an AIS with an ice volume of 22.3 msle compared to PI (Abe-Ouchi et al., 2015). This ice volume largely overestimates the AIS volume change inferred from the latest studies (Simms et al., 2019). It is clear that a significant larger AIS will create a colder and drier environment than a smaller ice sheet. In order to compare with PMIP3 results, the first preliminary results of PMIP4 are forced with the same AIS LGM configuration (Kageyama et al., 2020). Nonetheless, given the fact that the latest studies point to a lower ice volume, new PMIP experiments should consider the effect of a fully advanced, but smaller AIS. The alternative would be to employ fully coupled ice-sheet--climate models to simulate both the LGM climatologies and the LGM ice sheets."

With deleted the sentence:

"A way to potentially test the plausibility of the employed climatic fields is to compare with ice proxies."

We agree with the reviewer's opinion that in order to compare the model output with ice proxies it is first necessary to have consistency between the employed LGM AIS for climate models and reconstructions.

2. The basal melting of the ice shelf is fixed to zero in the simulations. I think this is a reasonable simplification to focus on the main topic of this study, however you should at least discuss the potential effect of the simplification you made. For example, while Obase et al. (2017, JCLIM) show that the basal melting at the LGM largely reduced compared with PI in their simulations with regional ocean model, the basal melt of LGM was still more than 50% of the PI experiment. Based on their estimates, the simulated area and volume in your experiment can be considered as the maximum estimate, and that the

uncertainties in the ice shelf basal melting can have an impact on the LGM AIS. Please add a discussion on this topic.

Indeed, adding basal melt would add a degree of difficulty. As shown in Golledge et al., (2012), oceanic forcing leads to a dynamic response of rapid ice streams, especially in the WAIS. Thus, including basal melt would most likely reduce the total ice volume and potentially affect the ice extension.

We found two studies that particularly address the problem of basal melting rates during the LGM, namely Obase et al., (2017) and Kusahara et al., (2015). Kusahara et al. (2015) used a fully advanced AIS to estimate the basal melting rates of ice shelves located at the border of the continental-shelf break. The basal-melting rates obtained were higher than PD values due to a greater exposure of the ice shelves to warm CDW. Nonetheless, these ice shelves are located at the continental-shelf break and these high melting rates do not necessarily apply in the interior of the continental shelf because of a more limited penetration of CDW. In our experimental setup the applied basal-melting rate refers to the interior of the continental shelf.

On the other hand, Obase et al. (2017) applied LGM conditions to an idealized PD configuration to investigate the response of basal-melt rates to a changing climate. However, as they pointed out, the changing basal mass balance actually modifies the thickness of the ice shelf and the positions of the grounding lines. This in turn affects the sea ice and the ocean around the ice shelves, which affects basal melting. Therefore, these melting values inside the continental-shelf are not valid for our experimental setup. We choose to set the basal-melting rates to zero to allow for a maximum ice extent.

We added two paragraphs in the Discussion section saying:
"In this study, no basal melting was considered during the LGM. Of course, this is a vast simplification of reality. Unfortunately, reconstructions of ocean subsurface temperatures at the LGM are not available, so that the geological evidence for basal melt is lacking. As shown in Golledge et al., (2012), oceanic forcing leads to a dynamic response of LGM ice streams in the WAIS. If basal melt would have been considered, this would have most likely reduced the total LGM ice volume and affected its extension. Thus, our

results represent an upper limit which would reduce when oceanic forcing is considered.

From the point of view of modelling, there have been some attempts to infer basal-melting rates. Kusahara et al., (2015) used a coupled ice-shelf-sea-ice-ocean model with a fixed LGM AIS extension, up to the continental-shelf break. In their model results, they obtained a larger basal melt value of ice shelves than PD. These large basal-melting rates occurred because the ice shelves were located at the edge of the continental-shelf break, where ice shelves are in contact with the warm CDW. However, these basal-melting values cannot be applied to the interior of the continental shelf as these waters do not penetrate so easily there.  On the other hand, Obase et al., (2017) simulated basal-melting rates on an idealized PD AIS to investigate the response of basal melt rate to a changing climate. However, these basal-melting rates are not realistic and cannot be applied directly to the AIS as the grounding-line advances during the LGM affect the climatic conditions and sub-shelf melting. In order to investigate the impact of realistic basal-melting rates it would be necessary to account for comprehensive parameterisations, such as PICO or PICOS (Reese et al., 2018; Pelle et al., 2019), or coupled ice-sheet-ocean models. This is out of the scope of this study. Furthermore, since our aim was to simulate a fully advanced AIS, as suggested by geomorphological records (The Raised Consortium, 2014), basal-melting rates were set to zero for the sake of simplicity in this work."

3. It is interesting to see that the differences in glacial atmospheric forcing caused large discrepancies in the simulated LGM AIS, especially for that of CNRM5. While it is not the main topic of this study to understand the cause of the difference in atmospheric forcing, I think it is valuable to discuss some possible reasons. For example,the result of CNRM5 reminds me of a study by Marzocchi and Jansen (2017, GRL)who compared the sea ice among PMIP3 LGM simulation. In their Fig. 3, you can find that CNRM5 simulates the smallest austral summer sea ice extent in LGM among PMIP3 models. This will cause warmer summer temperature over the marginal region of AIS and contribute to the negative mass balance. Perhaps, you may add one or two sentences on this point.

The CNRM5 model simulates the warmest LGM temperature, not only for the Antarctic domain, but this was also found in the NH (Niu et al., 2019). Our new results show even a smaller AIS due to the abnormal presence of ablation (Supplementary Material). In fact, in Kageyama et al., (2020) this model is represented as an outlier. We did not investigate the possible reasons for this, but truly Fig. 3 from Marzocchi and Jansen (2017) could hint to a potential explanation for these warm temperatures.

We added in the Discussion section:
"Nonetheless, it seems unrealistic that air temperatures were high enough to produce ablation during the LGM as seen in CNRM-CM5. The model CNRM-CM5 simulates the warmest LGM temperatures not only in the SH, but it has been also shown to simulate the lowest LGM volumes for the NH (Niu et al., 2019). A potential explanation for this behaviour can be due to sea-ice formation. As shown in Marzocchi and Jansen (2017), the CNRM-CM5 model simulates the lowest austral sea-ice extent. Such a low extent would increase surface temperatures through sea-ice albedo feedback. Hence, this could point to sea-ice formation as a crucial element in driving fully LGM conditions."

Specific comments:
P2L30-31: This sentence describes several processes, which affect the estimate of the volume of LGM AIS. However it is unclear how the modifications affect the estimate.Please add some explanations on this point. You may focus on one or two processes,which are relevant to this study.

The new sentence reads now:
"Whereas older studies estimated large sea-level contributions generally above 15 m (e.g. Nakada et al. (2000);Huybrechts (2002); Peltier and Fairbanks (2006); Philippon et al. (2006); Bassett et al. (2007)), more recent modelling studies and reconstructions have lowered these estimates to 7.5-13.5 m (Mackintosh et al., 2011; Whitehouse et al., 2012a; Golledge et al., 2012, 2014; Gomez et al., 2013; Argus et al., 2014b; Briggs et al., 2014; Maris et al., 2014; Sutter et al., 2019). This lowering in ice volume can be explained by the fact that the first ice-sheet models were based purely on the Shallow Ice Approximation (SIA) for inland ice. This solution solves for slow moving ice, based on shear deformation. However, later models include more sophisticated approximations (e.g. Shallow Shelf

Approximation, Full Stokes) with a better representation of fast flowing ice streams. These fast flowing regions contribute to a decrease in ice volume. Nevertheless, the latest LGM AIS volume estimates still differ by more than 5 msle. Part of this difference can be explained by spatial resolution and sub-grid scale grounding-line treatment (e.g. Goelzer et al. 2017; Pattyn 2018). Other possible explanations include the implementation of external processes, like the GIA (e.g., Whitehouse et al., 2019), or, as this work, the effect of uncertain climatologies and ice-sheet dynamics."

P3L3: I mostly agree with this sentence, but is it really true that the basal melting is negligible during LGM? For example, Obase et al. (2017, JCLIM) showed with regional ocean model that there is still some basal melting occurring at LGM. Please modify this sentence in a more modest way.

This issue was discussed above. The new sentence reads now:
"Ablation rates at the PD are almost zero except for localized areas (van Wessem et al., 2016, 2018). Because the LGM is a colder period, around 10 degrees as shown by ice core records (Jouzel et al., 2007), ablation rates in the LGM would have been probably negligible. On the other hand, basal melting rates from the LGM are difficult to estimate due to the scarcity of oceanic temperature reconstructions. Nonetheless, geomorphological records point to a fully advanced AIS during the LGM (The Raised Consortium, 2014). This could hint to low basal-melting rates inside the continental-shelf break."

P4L10: This sentence is difficult to read. Do you mean that in some models, the simulated results largely differ from ice core reconstructions? Please modify this sentence.

Yes, that is what we meant to say. The new sentence reads

"However, in some models, the simulated results differ from ice core reconstructions (Cauquoin et al., 2015). This may lead to an unrealistic configuration and thus it is necessary to evaluate the accuracy of model outputs."

P6L15-20: I had difficulty understanding this sentence, since I'm not familiar with an ice sheet model. Please describe this sentence in more detail. Why

do you use PD temperature field at sea level rather than surface? How does RACMO calculate the sea level temperature field? Do they assume a constant lapse rate in converting the temperature? If so, is the value of the lapse rate identical to what you chose in your ice sheet model?
P6L19: How did you decide this value of the lapse rate? Do you have any reference for this?

RACMO does not compute temperatures at sea-level, but at the surface. Yelmo, as well as other ice-sheet models, uses a lapse rate to scale the temperatures down to sea level and then scale them back up to the simulated surface elevation, to take into account changes in temperature and precipitation due to the elevation. Hence, Yelmo needs the surface temperatures simulated by RACMO as well as the PD surface elevation, to convert these temperatures to sea level. The same occurs for the LGM climatologies:  the LGM surface elevations provided by Abe-Ouchi et al., (2015) are needed to correct the climatologies with the elevation

The lapse rate value is an imposed value in ice-sheet models. It is not a uniform value over the whole Antarctic domain, but ranges from 0.015 K/m in most interior regions to 0.005K/m in the coastal zones (Fortuin and Oerlemans, 1990). Ice-sheet models commonly set this value to 0.008 K/m over the whole continent for simplicity (DeConto and Pollard, 2016; Quiquet et al.,2018; Albrecht et al., 2020). It accounts for the fact that changes in surface elevation imply also a change in temperature (colder temperatures at higher elevations). In order to improve the employed methodology we take into account changes in humidity by imposing two values, one for summer and another for annual temperatures.

The new paragraph reads:

"We apply a lapse rate correction that accounts for LGM minus PD changes in elevation (0.008K m$^{-1}$for annual temperatures and 0.0065K m$^{-1}$for summer temperatures) in concordance with other ice-sheet models (Ritz et al., 1997, DeConto and Pollard, 2016; Quiquet et al.,2018; Albrecht et al., 2020)."

P8L14: Are these results consistent with previous studies?

Yes, these results are similar to the basal sliding map of Golledge et al. (2012). We added the sentence
"These zones of fast flowing areas are similar to the predicted occurence of basal sliding from Golledge et al. (2012)."

P9L6: How did you define ground line temperature? Does the location of grounding line depend on simulations?

The grounding-line temperature is defined as the mean temperature of the ice column of all grounding-line points. The location of the grounding line is defined in Yelmo through the flotation criterium, hence it is different for each simulation.

The new sentence reads:
"Further inspection allows us to identify the atmospheric temperature close to the grounding line (Fig 7d) as a critical factor in determining how far the AIS advances. The grounding-line temperature is defined as the mean temperature of the ice column for all the grounding-line grid points."

P11L20-25: Please add a discussion on the role of basal melting in this subsection.
Done, we added there the paragraph of General Comment #2.

P12L1-2: I like this finding.
Thank you. As suggested by another reviewer we have highlighted this finding in the abstract.

P12L4-5: I don't think it's that simple. On one hand, the very thick and extensive PMIP3 LGM ice sheet can induce a drastic expansion of LGM AIS due to the large decrease in surface air temperature. However, on the other hand, the thick ice sheet will reduce the amount of precipitation, which will cause a thinning of LGM AIS, opposite to PMIP3 LGM ice sheet.
Indeed, a thicker ice sheet will tend to produce a colder and drier climate which can hamper the formation of a large ice sheet. We removed this last part of the article and finished with the discussion about the role of the employed LGM AIS in the CMIP3 experiments.

P12L8: You may cite a recent article by Kageyama et al. (2020, Climate Past Discussion), which discusses preliminary results of PMIP4 LGM experiments. Done, thanks for pointing this work out.

Fig. 5: It's hard to see the contour of the surface topography. Please modify this figure. Done

FIg.S1: I think this figure contains some important information on the reproducibly of modern Antarctic ice sheet. Please move it to the main manuscript Done

[revised manuscript text omitted]

---

## Author Response (AR2)

======== Editor comment =============

Comments to the Author:
Dear Javier Blasco and co-authors,

thank you for providing a revised version for your manuscript. I find that you adequately considered the questions and comments of the reviewers adequately in your reply and revised the manuscript accordingly. I am therefore accepting your manuscript for publication in TC. Nevertheless, I have a few minor technical comments which I ask you to consider before uploading the final version (see below).

We are very grateful to the Editor for accepting our manuscript for publication. Below you can find our response to each comment.

p3l6: 10 degree -> 10 °C
Done

p11l3: 2000 m: clarify if you mean meter below sea level
Done

p12l23: edit in LaTeX to: ice shelf--sea ice—ocean
Done

p14l2: add „a": via a subgrid
Done

p15l16: resultion -> resolutions
We could not find this error in the latest manuscript.

SM:

Fig. 1 & 2: please move numbers below every sub-panel down a bit to not overlap with the figure, for better readibility.
Done

Fig. 8: Why do you indicate CNRM-CM5 mass loss only as <0 but not with the actual value?
We extended the limits from -1 to 1 m/yr.

Please check the manuscript and the supplement for cases, where you miss a blank between a digit and the unit, e.g. 16km should be 16 km.

Done

[revised manuscript text omitted]